# The dynamics of prebiotic take-off: the transfer of functional RNA communities from mineral surfaces to vesicles
Dániel Vörös [1,2,3], Tamás Czárán [1] ✉, András Szilágyi[1,3] & Balázs Könnyű[1,3]

In this study, we propose a two-phase scenario for the origin of the first protocellular form of life, linking two RNA-world models by an explicit dynamical interface that simulates the transition of a metabolically cooperating RNA-replicator community from a mineral surface into a population of membrane vesicles. The two agent-based models: the Metabolically Coupled Replicator System (MCRS) and the Stochastic Corrector Model (SCM), are built on principles of systems chemistry, molecular biology, ecology and evolutionary biology. We show that the MCRS is easier to initiate from random RNA communities, while the SCM is more efficient at reducing the genetic assortment load during system growth and preadapted to later evolutionary transitions like chromosome formation, suggesting the former as a stepping stone to the later, protocellular stage. The switching between the two scenarios is shown to be dynamically feasible under a wide range of the parameter space of the merged model, allowing for the emergence of complex cooperative behaviours in metabolically coupled communities of RNA enzymes.

Students of the origin of life usually approach their topic either from a theoretical perspective focusing on the rules of self-organisation, or they attempt the synthesis of biomolecules in a more empirical setting[1–4]. An ideal, and certainly more fruitful approach would be one in which theoretical studies would instruct empirical work, and experiments would channel theory. The Miller–Urey experiment[5] was the first empirical breakthrough in studying the origins of life, creating biomolecules from inorganic building blocks using external energy sources. Following Miller's seminal experiment, several research groups attempted to unravel non-catalysed synthetic pathways for producing organic molecules under historically accurate (or at least so presumed) primitive Earth conditions[1,4,6–8]. While some of these pathways may be operational in the lab, they require specific conditions and high concentrations of reactants to produce a sufficient yield of biogenic compounds instead of a tar-like mixture of substances[9,10], the generic outcome of such non-enzymatic, stoichiometrically and kinetically uncoordinated reaction networks. The presence of inorganic catalysts, such as metal ions and mineral surfaces, could have channelled the reactions to some extent to yield biologically relevant molecules, enabling previously unlikely catalytic routes to become feasible[7,11–16]. Yet, as such inorganic catalysts lack catalytic specificity, side reactions may still have hampered the maintenance of a persistent reaction network. To ensure that source matter is transformed into biotic target molecules, a controlled environment and specific (bio-) catalysts are required. The catalytic machinery of recent organisms is highly complex and specialised, involving many different types of biomolecules and inorganic cofactors. It is unlikely that this complexity could have spontaneously emerged without the aid of substrate-specific catalysts.

The most likely candidate for prebiotic enzyme functions is RNA which still acts in each recent living cell in different catalytic roles. The RNA world hypothesis[17] posits that the first RNA molecules, while encoding the information necessary for their replication in their sequences might have also acted as catalysts due to their secondary and tertiary structures and reactive nucleotide sidechains. This dual role (genetic and catalytic) makes RNA a suitable candidate for playing the protagonist role in the theatre of the origin of life. RNA enzymes (ribozymes) are capable of catalysing a wide range of reactions[18–24], and they can be replicated, allowing evolution to act on them to select for the assembly of a suitable set of catalysts channelling a reaction network that, in turn, supplies building blocks for their replication. Although abiotic synthetic pathways in solutions may have produced the essential components of RNA nucleotides (ribose and RNA nucleobases[4–6,8]), in a dilute bulk water phase the degree of RNA polymerisation (i.e., the average length of RNA polymers produced) is low, partly because hydrolysis is thermodynamically more favourable than condensation in terms of entropy gain[12]. Mineral surfaces (like those of montmorillonite)[12,14] or eutectic solutions[25,26] or the inner space of dehydrated lipid vesicles[27] may have provided a better environment for the formation of RNA molecules by increasing the effective concentrations of nucleotides. In addition, mineral

[1]HUN-REN Centre for Ecological Research, Institute of Evolution, Budapest, Hungary. [2]ELTE Eötvös Loránd University, Institute of Biology, Budapest, Hungary. [3]Parmenides Foundation, Pöcking, Germany. ✉e-mail: czaran.tamas@ecolres.hu

surfaces may have protected RNA against UV radiation[28] that breaks RNA polymers by reducing the degrees of freedom of movement of the broken parts[29,30], thus enhancing spontaneous repair and supporting the maintenance of longer polymers[12,14] and catalytic capabilities[31,32]. We believe that mineral surfaces provide a more plausible scenario for the origin of RNAs than bulk water phases[33–35], although there is no unequivocal scientific consensus on this matter. We assume that a diverse set of RNA enzymes may also have emerged on mineral surfaces[16,31,36–38] which could have booted up self-sustaining replicator communities based on metabolic networks collectively catalysed by the community itself[36,39–41]. The metabolically active replicator community becomes self-sustaining by supporting the replication of its members. Through this positive dynamic feedback, such communities could become subjects of Darwinian evolution towards more efficient metabolism and faster replication.

In a prebiotic community of primitive replicators lacking the sophisticated genetic regulatory mechanisms at the disposal of recent cells, the coexistence and cooperation of different types of (RNA) replicators is far from trivial[42]. Even if an advantageous set of ribozymes could emerge, the community as a functional unit would still be at risk of extinction due to either demographic stochasticity[43–46], parasite invasions[43–45] or competitive exclusion by faster-replicating RNA replicators[39,47]. The survival of replicator communities requires either some form of spatiotemporal structure[39,45,48] or compartmentalisation[34,49–51], with functionally complete communities enjoying a selective group advantage[52]. Mineral surfaces provide limited diffusion for the molecules attached to them, resulting in a vague spatial segregation mechanism allowing for different selective regimes to act at localities harbouring different replicator assemblies. This kind of local dynamics can maintain the information content (enzymatic diversity) of metabolically complete replicator patches[40,41,53–55].

Although mineral surfaces provide a more suitable environmental context for the origin of RNAs than bulk-phase solutions for biochemical and dynamical reasons, their efficiency can be enhanced many times over by allowing the emergence of a new level of replicator community organisation on which selection could act. On a continuous 2D surface, the separation of local replicator communities is only partial due to the limited diffusion of replicators in two ways. First, established, efficient communities can be dissolved as cooperating replicator types stochastically drift away from each other. Second, invading non-enzymatic or parasitic replicators from more distant sites may destroy established local communities[45,56]. According to theoretical studies, the most effective way for selection to act on primitive replicator sets is the compartmentalisation of replicators inside well-defined vesicle-like structures[34,50,57,58]. This introduces two distinct levels of selection on the replicators: (1) inside each vesicle all replicators cooperate to increase group (vesicle) fitness and compete with each other for reproduction, (2) and competition among the groups of replicators contained in the vesicles. On this note, a seemingly reasonable argument against the hypothesis of a surface-bound origin for life may be that prebiotic replicator communities *ab ovo* wrapped into vesicles had a selective advantage over those evolving on a mineral surface, rendering the surface-bound scenario superfluous. The problem with this argument is that a viable compartment-bound replicator system (like e.g., in refs. 59–63) cannot be assembled and maintained without rigorous stoichiometric and/or kinetic coordination between the growth of the replicator system and that of the vesicles wrapping them, as required in Gánti's Chemoton[64]. This means that it has to be the replicator community itself that produces the lipid molecules making up the vesicles at a speed precisely corresponding to their own replication rate, otherwise the replicator population either grows relatively too fast and bursts the vesicle wrapping it or grows too slow and becomes diluted over time. The surface-bound metabolic ribozyme community scenario allows for the gradual evolution of a lipid-producing ribozyme[41,65] set with consistent selection advantages during the transition phase without the need to assume the miraculous, abrupt emergence of a functional metabolic replicator set that also produces lipids in a coordinated manner for the growth of its vesicles. The advantage of the surface is twofold in this scenario: it provides the spatial structure that allows for the coexistence of cooperating metabolic

replicators and allows the metabolic replicator system to evolve (through mutations and selection) the necessary ribozymes for the coordinated production of the membrane constituents. Bulk phase dynamics would not deliver any one of these possibilities.

If we accept the feasibility of the role of mineral surfaces in the origin of life, a clear corollary is that surface-bound 2D replicator communities had to move into 3D vesicles at some stage of prebiotic evolution, because modern life is uniformly cellular. Could the transition between two radically different environments (mineral surfaces and lipid vesicles) be possible through feasible ecological and evolutionary dynamics? The prebiotic synthesis of vesicle building blocks (e.g., lipids) is shown to be possible, as is their spontaneous organisation into vesicles both in water and on surfaces[66–75]. There are hypothetical encapsulation mechanisms[33,59,60], some with experimental support, such as the wetting-dewetting cycles of vesicles[76] or wet-dry[51]/freeze-thaw cycles[77], which can be assumed to support the transfer of surface-bound replicator communities into vesicles once the coordinated growth of replicator populations and vesicle compartments has evolved. This environmental change does not have to affect the enzymatic properties of the ribozymes[28,38] to a large extent, and membrane vesicles could even facilitate the replication of oligomers[61]. Even so, the question remains whether the transferred metabolic communities could survive the 2D-3D transition that changes the selective regime around them regarding community topology, functional diversity, stability and enzymatic properties.

The present study aims to investigate the ecological and evolutionary criteria of the transition, focusing on changes in the dynamics and the organisation of the replicator community during its take-off from the surface into the vesicles. One of the crucial empirical results aligned with our hypothesis on the mechanism of a prebiotic take-off is the exciting work of Gözen[76], who shows that the actual departure of membrane vesicles from a 2D surface is physically feasible once the appropriate lipid molecules are present. This is the scenario that we consider to represent the transition from the surface to the vesicle regime.

The two cornerstones of our theoretical approach are the Metabolically Coupled Replicator System (MCRS model[39,54]) corresponding to a supposed earlier stage of prebiotic evolution on a 2D mineral surface and a subsequent, still primitive protocellular system represented by the Stochastic Corrector Model (SCM[50,78,79]). In the MCRS model, cooperative replicator types provide catalytic support to a common metabolic function, supplying the necessary monomers for the replication of all members of the replicator community (see Fig. 1). Metabolism works only with a complete set of enzymatic functions present. This concept works in a surface-bound context due to limited metabolite diffusion (assuming the products of metabolism cannot disperse far on the surface), while in a well-mixed bulk phase (represented by the mean-field version of the model) the system collapses as the fastest-replicating replicator species competitively excludes the other types, rendering metabolism of the system incomplete[39]. In the surface-bound system, spatial patches with a balanced content of replicator types have a selective advantage, ensuring the system's survival. The (weak) group selection acting on the local communities of ribozymes[80,81] guarantees that the MCRS is robust on both the ecological and the evolutionary timescales. Assuming increasingly realistic chemical details, later versions of the MCRS model have become ever more robust[39,54,58,80,82–86].

The Stochastic Corrector Model (SCM)[50,58] represents the protocellular dynamics of replicators encapsulated in vesicles. The replicator content of the vesicles is assumed to increase by replication and to be redistributed in two daughter vesicles upon fission, with a strong stochastic drift (assortment load of replicators) ensuring the random reassembly of metabolically active communities. Group selection[52], which guarantees the ecological and evolutionary stability of the system[50,78,79], acts at the level of the vesicles. We have developed an MCRS and an SCM model assuming the same replicator dynamics in the two, to appropriately investigate the effect of transferring replicator samples from the MCRS (2D) to the SCM (3D) environment while keeping the cooperative dynamics intact.

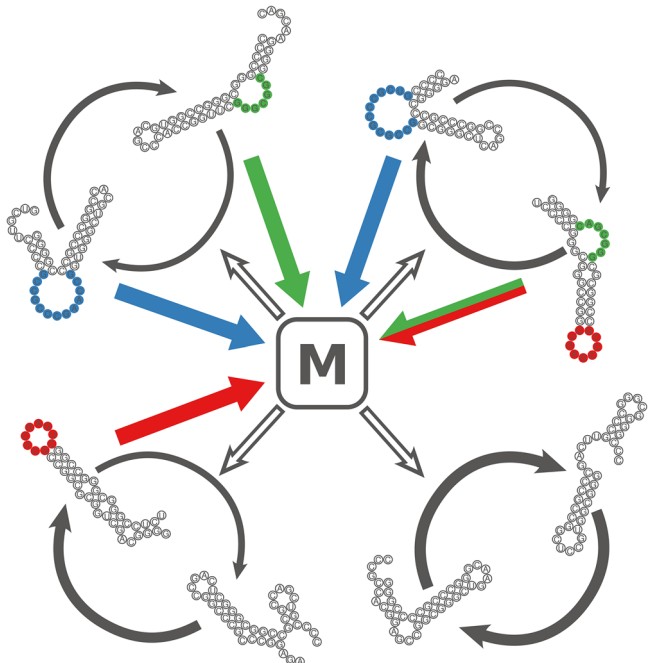

**Fig. 1 | Metabolic cooperation among replicator types.** Replicators are RNA sequences with secondary structures that may have catalytically active sites (coloured bases). Three types of enzymatic activities are assumed to contribute to the common metabolic function $M$ (red, green and blue arrows); in exchange, the metabolic function facilitates (hollow grey arrows) the replication of all replicators (grey arrows). A replicator and its complement copy can be categorised by the set and the topology of enzymatic activities they harbour. Classic specialist replicators (bottom left pair) have exactly one catalytic site on one strand. Their complementary strands are considered parasites because they do not contribute to the common metabolism, although they are essential templates of the specialist enzyme. Real parasites are replicators neither of the complementary strands of which has enzymatic activity (bottom right pair). Trans-promiscuous replicators (top left) have identical or different activities on both strands. Cis-promiscuous replicators accommodate multiple catalytic sites on the same strand—their complementary strands may or may not be catalytically active (top right pair).

The two models used in this study are similar to the original ones, except that we have tailored them to be consistently compatible for comparing their dynamical outcomes before, during and after the transition. This includes a substantial change in the SCM to handle mortality events on the vesicle level, which was not the case in earlier SCM versions. Now, the two models are comparable in the discrete events on the replicator level. Although both model concepts have been studied previously, the real innovation in the approach taken here is in studying the transitory dynamics between the two different selection regimes by linking—both technically and conceptually—the MCRS and the SCM frameworks. This is feasible because the two different scenarios are now algorithmically possible to link. By transferring replicator patches from the surface to vesicles we show that although the evolution of replicator traits and the dynamics of community assembly differ in these two spatial contexts, both can maintain ecologically and evolutionarily stable populations, resisting competitive exclusion and parasite invasion.

## Results

We have analysed the behaviour of the two models separately, with special emphasis on the transition between them. Since in any well-mixed (mean-field) context the Metabolically Coupled Replicator System (Fig. 1) is doomed to extinction due to competitive exclusion[39,47,87], group selection[52] imposed on the replicators by surface dynamics and compartmentalisation must be the key to ensure the coexistence of all the necessary replicator types

by shifting the focus of selection from individual replicators to vaguely (in MCRS) or strictly (in SCM) defined local communities. If group selection works properly, the fitness advantage of groups overrides the selective loss in individual replicator fitnesses. To be a feasible representation of the emergence of life, any model of replicator cooperation should satisfy the following conditions: (1) ecological stability: the frequencies of all replicator types should attain a non-zero attractor in each timeframe; (2) robustness: the model's behaviour should be similar over a wide range of parameters and withstand the effects of perturbations; and (3) evolvability: replicators and replicator communities should be allowed to change (e.g., by mutation) and be selected for new properties improving the overall fitness of the community. The analyses of the results were conducted with these key points in focus.

### The surface-bound stage: Metabolically Coupled Replicator System

Simulations were initiated from an unevolved, but enzymatically active (initial) replicator pool (as specified in the "Methods" section). In principle, a single complete metabolic neighbourhood (a "seed" locality) in the system is sufficient to start the simulation. Of course, on a practically infinite lattice/surface over a very long time period, the random assembly of such a metabolically complete neighbourhood can be taken almost for granted, but on a relatively small grid and limited simulation time permitted by the computational requirements of following the dynamics of the system, we needed to accelerate this process (hence the composition of the random pool, see "Methods"). Even then, only a few seed localities occur in the initial lattice configurations, so that after initialisation the grid occupancy drops, leaving an almost empty lattice, save for a few metabolically complete, therefore successful and expanding neighbourhoods. After this transient phase, the number of replicators increases and reaches a quasi-stationary grid occupancy of around 80%, almost independently of the actual parameter setting. A systematic screening along feasible ranges of the spatial parameters such as the sizes of neighbourhoods ($N_{met}$, $N_{rep}$, see "Methods"), replicator diffusion ($D$) and system size (the number of types of enzymatic activities, $A$) has been conducted in order to find parts of the parameter space in which the MCRS system is viable. In accordance with our previous works, the system showed a high robustness in response to these changes[54]. The most important decisive factor turned out to be the size of the metabolic neighbourhood ($N_{met}$): at too low $N_{met}$ values the metabolic neighbourhood was too small to include a complete metabolic community, while at too high values (huge metabolic neighbourhood sizes) the system approaches the mean-field model (see ref. 39), reducing the advantage of rare replicators that results in the disappearance of replicators with low replication rates and eventually the collapse of metabolic community (see Fig. 2). The lowest critical metabolic neighbourhood size depends on system size (Fig. 2): only larger metabolic neighbourhoods can accommodate an increased number of different enzymatic activities. Larger system size ($A$) could be easily compensated with a slight increase in $N_{met}$ within the range of $A \in [3, 5]$; above $A = 6$, larger increases in metabolic neighbourhood size are needed to ensure that each replicator type is represented by at least one copy in a sufficient number of neighbourhoods. Higher $N_{met}$ simulations had shorter transient phases, probably due to the decreased variance of larger neighbourhood configurations.

The size of replication neighbourhoods ($N_{rep}$) and diffusion ($D$) have similar effects on the model's behaviour: increasing any one of them provides more spatial mixing for replicators. Mixing, on the one hand, helps replicators to get into metabolically complete neighbourhoods in which they are not surrounded mostly by their own copies, but on the other hand, intensive mixing supports the invasion of parasites as well. We have found that increasing the system size $A$ requires more intensive spatial mixing, but too much mixing (which is quite unrealistic to assume for RNA molecules anchoring on mineral surfaces) may be detrimental to system survival, so we used $D \in \{0, 4\}$ and $N_{rep} \leq N_{met}$. We have conducted simulations at different system sizes ($A$), diffusion ($D$) and sizes of neighbourhoods ($N_{met}$, $N_{rep}$) (see Fig. 2, Table 1); we present the 2D-3D transition of replicator

**Fig. 2 | Parameter mapping of MCRS simulations.** Scanning for the viable part of the parameter space in the MCRS, showing the proportion of ten parallel simulations with replicators still alive after 2000 timesteps and the proportion of parasitic replicators in the last generation. The fill colour of the squares represents the frequency of simulations still alive after 2000 generations initiated with different random seeds (dark blue: all extinct, light blue: all survived). The diameter of red dots inside the squares represents the mean of relative parasite frequencies at the end of surviving simulations. For simplicity, in these simulations $N_{met} = N_{rep}$. Each data point represents the average of 10 repeats.

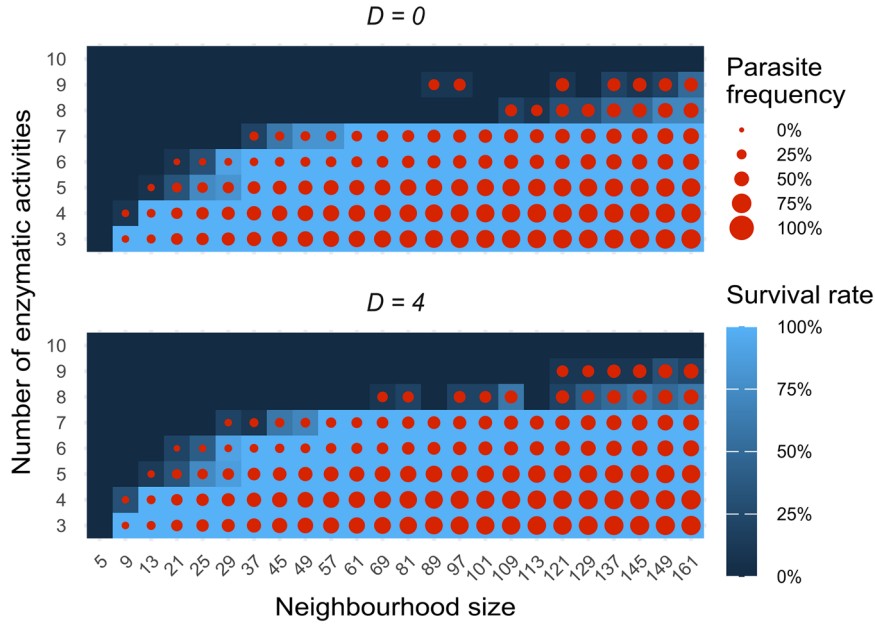

## Table 1 | Parameters and their values of the models

| Parameter | Description | Value |
|---|---|---|
| $A$ | Number of enzymatic activities for the metabolism (system size) | **3–10** |
| $p_{sub}$ | Per-base probability of substitution | 0.05 |
| $p_{ins}$ | Per-base probability of insertion | 0.005 |
| $p_{del}$ | Per-base probability of deletion | 0.005 |
| $E_{min}$ | Empirical minimal folding energy | −25 kcal/mol |
| $c$ | Scaling parameter mapping $\Delta G_{min} = E_{min}$ to $P_{fold} = 1$ | 0.3 |
| $d$ | Scaling parameter determining the shape of $\Delta G_{min} - P_{deg}$ function | 0.2 |
| $g$ | Replication rate scaling parameter | 10 |
| $l$ | Replication rate scaling parameter | 1.0 |
| $b_1$ | Time demand of replication initialisation | 0.75 |
| $b_2$ | Per-base time demand of strand elongation during replication | 0.005 |
| $\xi$ | Additional GC bases' contribution to enzymatic activities | −0.3 |
| $\sigma$ | Strength of sub-additive nature of cis-promiscuous active sites | 1.1 |
| $Z$ | The number of columns and rows of the grid | 300 |
| $N_{met}$ | Size of metabolic neighbourhood | **5–161** |
| $N_{repl}$ | Size of replication neighbourhood | **5–161** |
| $D$ | Strength of replicator diffusion | **0, 4** |
| $C_e$ | Claim of the empty site to remain empty | 0.1 |
| $N$ | Number of compartments | **250, 500, 750, (1000)** |
| $S$ | Split size | **50–100 (90)** |

The first section contains general parameters corresponding to replicator functions, while the second and third sections hold the special parameters of MCRS and SCM dynamics, respectively. Parameter ranges with more than a single value applied are indicated in bold numbers. Evolving traits of the replicators can be found in Supplementary Table 1.

community with the smallest metabolic neighbourhood that can sustain the system with 7 enzymatic activities: ($A = 7$, $N_{met} = 57$, $N_{rep} = 49$ and $D = 4$).

Due to strong demographic and functional constraints acting on the replicator communities of the MCRS, the simultaneous evolution of replicator structures is, in effect, a tightly self-regulated coevolutionary process both in terms of the division of labour between different replicators and the trade-offs between different traits of the same replicator. All viable simulations end up in a unique distribution of replicator types but without any conspicuous spatial pattern like distinct patches or spiral waves emerging. The actual outcomes depend on environmental effects like the diffusibility of the replicators ($D$, $N_{rep}$) and the metabolites ($N_{met}$) on the surface, but also on "historical" contingencies influencing the trajectories and the stationary states of the coevolutionary process such as the actual sequence of functionally important mutations occurring in the replicator community, which explains the differences among replicate runs of the simulations. Below, we focus on the common features of the coevolutionary dynamics unfolding in the MCRS.

The time plots of replicator frequencies, average replicabilities ($R$), lengths, foldedness (Gibbs free energy), and catalytic activities of the replicators in the MCRS are shown on the left column of Fig. 3. Fuelled by the weak group selection on local replicator communities, all these traits evolve towards quasi-stationary distributions, ultimately ensuring that all the replicator types present have the exact same (zero) relative fitness (a necessary condition if they are to coexist in a quasi-steady state) and that the common metabolism remains functional, despite parasitic replicators being always present at the dynamic equilibrium of loss by selection, reoccurrence by mutations and spread by replicator diffusion. The MCRS system selects replicator classes by length and catalytic functions, minimising the lengths of the sequences retaining the necessary catalytic structures, thus maximising catalytic cooperation, so that metabolic neighbourhoods can still contain replicators replicating as fast as possible while collectively providing all the necessary secondary catalytic structures responsible for a complete local metabolism (see Fig. 3). Recall that length is an important determinant of replication rate $R$. The zero-relative-fitness criterion requires that the advantage in replication rate of shorter enzymatically functional sequences (like type 1 and type 7 in the MCRS simulation of Fig. 3) be compensated by an increase in the net catalytic help they provide to longer, slower replicating types (like type 2 and type 4 in the MCRS simulation of Fig. 3). This

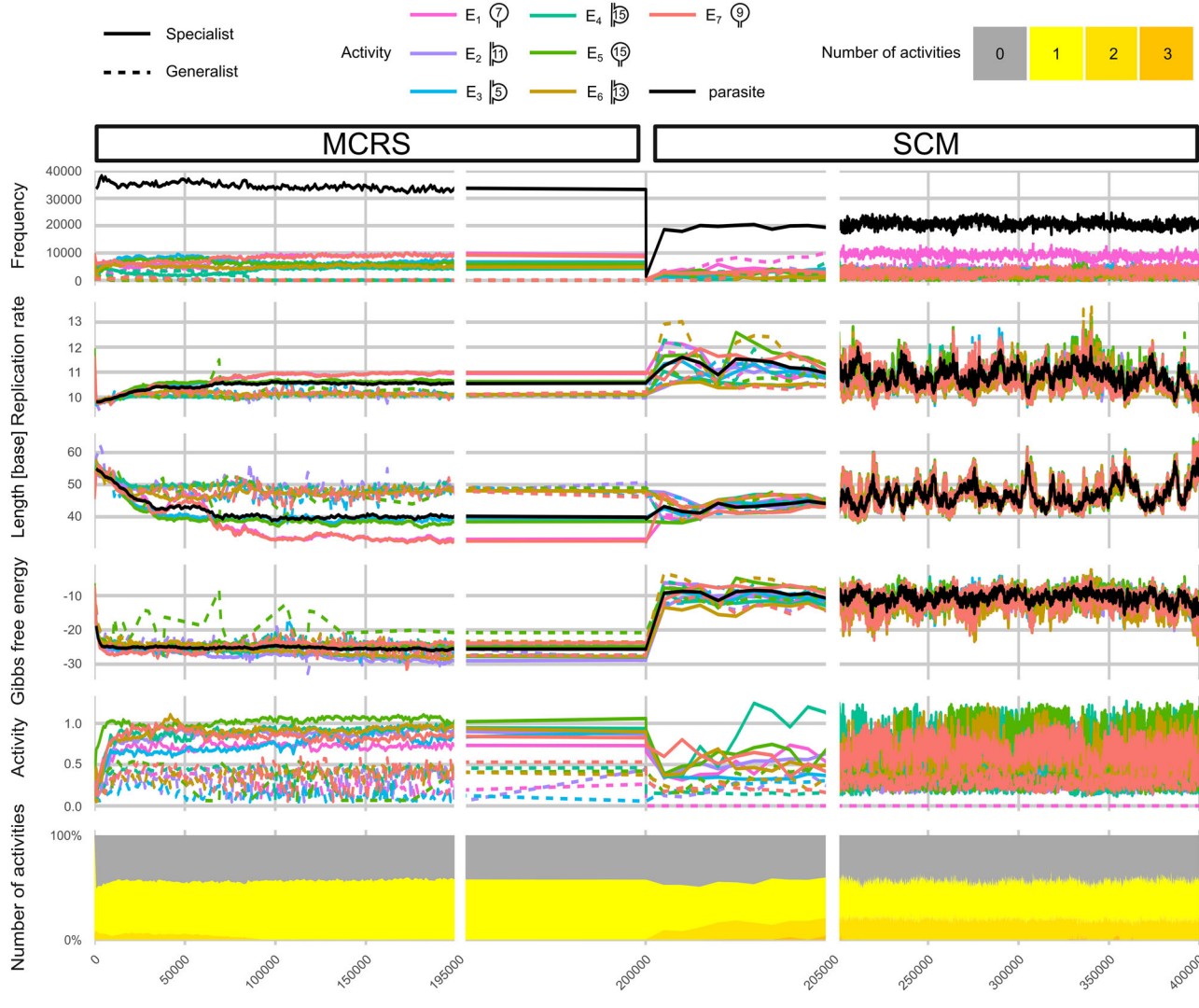

**Fig. 3 | The temporal change of replicator traits before, during, and after the transition from MCRS to SCM.** Replicator types are categorised by their catalysed enzymatic activities: replicators with a single activity (specialists) are represented by solid lines (line colour corresponds to the catalysed activity), while cis-promiscuous replicators by dashed lines (generalists). In the case of cis-promiscuous types, a replicator is represented in all the corresponding activities (e.g., a $E_{1,2}$ replicator's minimal free energy contributes to both dashed lines corresponding to activities 1 and 2 in the 4th row of the figure). In the legend, diagrams indicate the 2D structure (loop or interloop) and size (number of bases) of enzymatic motifs. Indicated replicator traits: relative frequency; mean of replicability ($R$); sequence length; minimal Gibbs free energy; enzymatic activity; and in the last row an area plot of relative frequencies of replicators with $A_n$ activities (coded by grey and yellow colours). At $t = 200,000$, a transition event occurred: the replicator content of 100 vesicles was sampled from the MCRS grid, and an SCM simulation was initiated with vesicles containing the replicator populations of the sample elements as indicated by the top panel labels. Note that the timescale is 20-fold zoomed in for the transition period. The relevant parameters for the MCRS are $N_{met} = 57, N_{rep} = 49, D = 4$; for the SCM: split size $S = 90$. During the whole simulation $A = 7$ was used.

compensatory catalytic effect is a typical example of self-regulation through relative frequency: faster-replicating ribozymes become common, so they provide more help to slower replicating low-frequency species within the coevolving community. Note that since pairs of strands (regardless of their catalytic activity pattern; see Fig. 4) are of the same length (disregarding insertion/deletion mutations during replication) the system needs to optimise the lengths and catalytic pattern contents among the different pairs of strands, i.e., among different replicator species. The Gibbs free energy levels of all the dominant secondary structures quickly converge towards the attainable minimum level (Fig. 3). As free energy contributes the most to the degradation rate ($P_{deg}$), it is safe to assume that the MCRS optimises for the lowest possible degradation, meaning that the system is 'degradation driven'. The high level of lattice occupancy (~80%) supports this conclusion. Variance in the strengths of enzymatic activities is low, but they form two well-distinguishable groups. As an emergent means of information

integration, replicators may acquire multiple catalytic activities on each or both of their complementary strands. This feature is called catalytic promiscuity. We have categorised the replicators by their level of cis-promiscuity (carrying multiple catalytic activities on the same strand), and trans-promiscuity (both complementary strands catalyse at least one metabolic reaction). As multiple catalytic sites may interfere with each other's activity, the mean activity of individual cis-promiscuous catalytic sites is lower than that of specialist (single-activity) ones (for more details see "Methods").

Regarding the effects of environmental variables on catalytic promiscuity patterns in the MCRS, the general result is that at restrictive (i.e., relatively small) metabolic neighbourhood sizes ($N_{met}$) some of the catalytically active replicators of the MCRS tend to evolve cis-promiscuity, but trans-promiscuity occurs far more frequently (Figs. 3, 4). The theoretical optimum solution for minimising the necessary metabolic neighbourhood

**Fig. 4 | The evolved states of catalytic activities in an MCRS simulation after 200,000 generations.** The distribution of replicator types present, categorised by their cis-promiscuity type and the number of their enzymatic activities on one strand (x-axis), against the same features of their theoretical (error-free) complementary sequences (y-axis) in a model snapshot. Cells represent the trans-promiscuity states of the corresponding complementary pairs of strands. Combinatorically valid cis-promiscuous types absent in the actual set of replicators are not shown (e.g., no $E_{2,4}$ replicator is present in this timeframe of the simulation, so the corresponding row and column are not shown). Colours indicate relative replicator frequencies on a log scale. Numbers inside the cells indicate the percentile frequencies (%) of the corresponding complementary pairs. Real parasites (neither strand is metabolically active) are represented at the bottom left cell. All other cells in the leftmost column and the bottom line represent replicators with only one of their complementary strands active metabolically—the inactive strands of these are functional parasites. Parameters of the simulation as in Fig. 3.

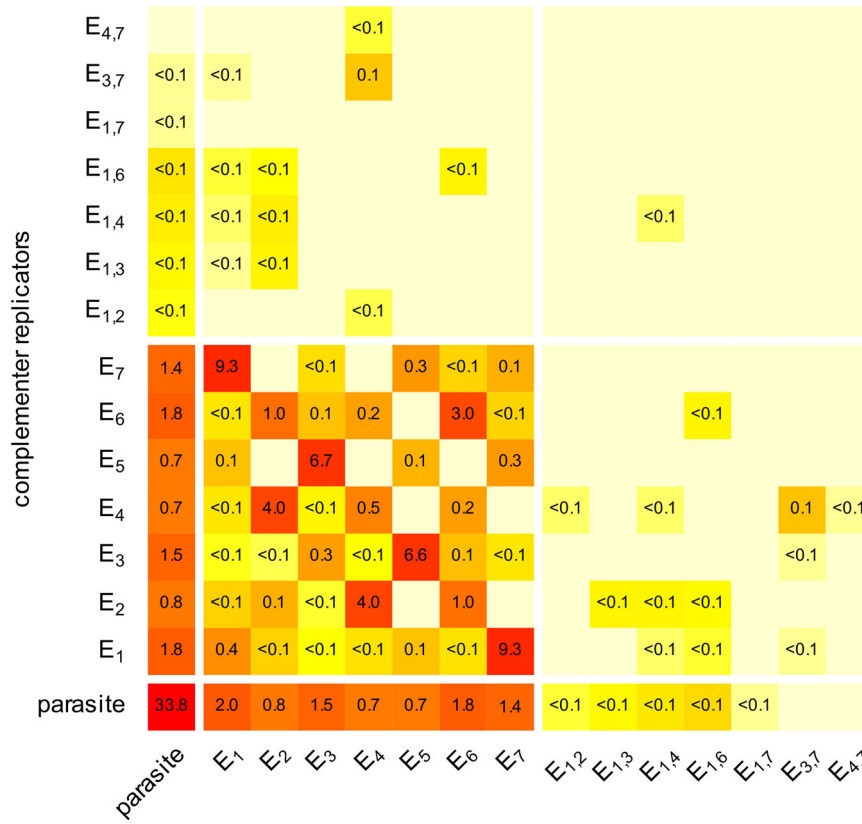

existing replicators

size while maintaining robust catalytic functionality in local metabolic communities is the evolution of cis- and trans-promiscuous replicator types (with multiple different catalytic active centres on both complementary strands). We do not see this happening in the MCRS simulations, which may be due to insufficient time given to coevolution to reach the optimum (being still in a transient), or to the interference of catalytic sites in cis-promiscuous replicators (see "Methods" for details) adversely affecting local metabolisms ($M$) even if they are present at high frequencies on the lattice. To answer the question of whether the quasi-stationary states of the MCRS simulations are still transient, another simulation was initiated with a cis-trans-promiscuous replicator accommodating 6 different catalytic activities on its complementary strands and found that this hyper-promiscuous pair of strands does not persist (see Supplementary Note 1). This supports the second hypothesis: it must be the competitive dominance of trans- over cis-promiscuous replicators within metabolic neighbourhoods that keeps cis-promiscuity rare. Yet, the high frequency of trans-promiscuous replicators in the simulations suggests that trans-promiscuity plays an important role in ensuring metabolic integrity within relatively small metabolic neighbourhoods in the MCRS.

**The transition between the two stages**

The algorithm modelling the process of prebiotic take-off, i.e., the transition from a surface-bound Metabolically Coupled Replicator System to a compartmentalised one comprises three procedures: (1) taking samples of the replicator community from the quasi-stationary state of the MCRS using circular sampling patches, (2) placing the samples into vesicles, and (3) letting the SCM continue to operate on the vesicle communities thus produced.

At sampling unit sizes consistent with the actual system size and metabolic neighbourhood size of the sampled MCRS simulation ($A = 7$, mean number of replicators per sample = 34.98, CV = 0.24), a relatively high, ~29.9% proportion of the vesicles turns out to be metabolically

complete. However, only ~6% of the vesicles is capable of spreading in long-term SCM simulations initiated with single vesicles formed by asynchronous transition (based on 5 replicate SCM simulations initiated with different random seeds for the 1000 vesicles sampled).

Initiation of the SCM with multiple different vesicles can be either sequential or simultaneous. In the sequential case the SCM is continuously injected by vesicles with replicators sampled from the surface during an extended sampling period (like in refs. 33,76), while in a simultaneous transition, the replicator content of multiple new vesicles is sampled from a single snapshot of the surface replicator community, i.e., it is a single event (more in line with wet-dry and freeze-thaw cycles[51,77]). In the sequential transition case, the momentary variance of replicator properties in the SCM increases during the sampling period, while the overall behaviour of the sampled quasi-stationary MCRS simulation does not significantly change (the sample size was 100 vesicles per generation, taken from the $300 \times 300$ grid of the MCRS). Due to the high replicator variance, the results are difficult to interpret, and the mapping of the viable part of the parameter space gets extremely slow. After the initiation period, the momentary variance of SCM replicator properties shrinks in both cases, indicating that the dynamics and the replicator content of the vesicle population converge. Since no substantial difference between the two cases could be observed, only simultaneous initiation by 100 vesicles was used in all subsequent simulations.

Even a short glance at Fig. 3 reveals that the two regimes (MCRS and SCM) are characteristically different. These differences are not the product of a bottleneck effect or the limited evolutionary capacity of either of the models, as shown in Supplementary Note 2, but rather the inherent differences between the dynamical contexts of the evolution of replicator traits along the two different pathways. After the transition the surviving vesicles quickly (in about 1500 generations) reach a fluctuating, but in the long run also quasi-stationary, state in terms of the trait distribution of their replicator communities.

## The protocellular stage: the Stochastic Corrector Model

The SCM could be successfully initialised with the same conditions and from the same unevolved but enzymatically active random pool of metabolic replicators as MCRS simulations within the range $A \in [3, 5]$, but not with system sizes $A > 5$. Thus, comparing the results of SCM started up from the random pool and SCM initialised from quasi-stationary MCRS simulations (see the "Transition" section) we conclude that SCM simulations with larger system sizes have a chance to persist only if initialised by vesicles sampled from MCRS simulations. This may be the consequence of the different catalytic activity patterns in the random pool and the evolved MCRS sample pool—mainly due to the lack of cis- and trans-promiscuity in the random pool, both of which efficiently reduce the assortment load that the SCM is known to be particularly sensitive to[88]. In this sense, we may assert that the SCM is unlikely to take off from a random pool of replicators, whereas it may emerge much easier from an MCRS-like community of replicators.

The differences between individual replication rates can be balanced by drift during split events in the distribution of replicator types within the daughter vesicles due to the relatively low number of replicators in the parent vesicle at division (split size $S$). This stochastic correction mechanism generates heritable variance in the metabolic fitness of the vesicles upon which selection may act, but it is effective only at relatively small split size $S$[50]. The ensuing non-equilibrium dynamics maintain the persistence of viable vesicles at the cost of a substantial assortment load. As split size increases, the distribution of replicator types in the daughter vesicles approaches that of the parent vesicle, with its variance approaching zero at the $S \to \infty$ limit. This limit represents the mean-field case known to weaken group selection and destroy metabolism due to competitive exclusion at the replicator level. Therefore, split size $S$ should be kept as low as possible to maintain stochastic correction. On the other hand, at larger system sizes $A$, simulations could not survive below a certain split size $S$, because random splitting too often produces vesicles missing at least one of the essential catalytic activities and thus incapable of producing monomers for replication, imposing assortment loads too high for a sufficient number of vesicles to survive. Exploring the parameter space suggested that, at $A = 7$, simulations initialised with vesicle contents sampled from an MCRS simulation required a split size of at least $S = 90$.

Another parameter of the SCM simulations on which system persistence depends is the number of vesicles $N$. After an initial screening with $N = \{250, 500, 750, 1000\}$, we found that the survival rate drastically drops at population sizes below $N = 750$, so for all subsequent simulations, we used $N = 1000$. This also ensures that the total number of replicators is comparable to the number of occupied cells in an MCRS simulation.

The coevolutionary process of the replicator communities in the SCM follows a pathway different from that in the MCRS. Group selection among the disjunct communities in vesicles is stronger than among more diffuse, overlapping surface neighbourhoods, which leads to more efficient selection, and, thus, faster changes both in replicator properties and community structure. Note that, due to the highly combinatorial nature of community assembly parallel SCM simulations with different random seeds, resulted in quite different results, yet their overall dynamical behaviour was similar, and the ensuing quasi-stationary states are ecologically stable. As a result of all this, the compositional diversity of the vesicle populations is also higher than the diversity of an MCRS lattice, as vesicles with nearly equal fitness (in terms of multiplication rate and survival of offspring vesicles) but with different replicator communities coexist without any mixing of content among vesicles. Evolved replicators in the SCM environment tend to have relatively high Gibbs free energy (Fig. 3), meaning that the secondary structures are less stable than in the MCRS. This implies higher degradation rates but also allows for faster replication. The stochastic correction effect that balances vesicle content is less effective at larger split sizes $S$ due to overproduction of replicators with high replication rates (see ref. 50).

The replicator transfer from the surface (MCRS) to vesicles (SCM) rarely yields viable replicator communities, thus the number of replicators decreases after the transition (see the middle column of Fig. 3). Also, there is a drop in enzymatic activities that can be attributed to two main factors. One is the need to reduce the assortment load in the SCM, favouring the selection of replicators with cis-promiscuous activities. Cis-promiscuity necessarily leads to lower catalytic activities because of the supposed structural interference of active sites on the same strand. The other is due to the higher average Gibbs free energy level of the replicators in the SCM: since their secondary structures are less stable, replicators spend less time in their catalytically active fold. This drop in enzymatic activities is corrected later to a small extent through the gradual optimisation (fine-tuning) of replicator lengths and secondary structures by the coevolutionary process to achieve the best possible catalytic performance. The increase is most striking in the case of cis-promiscuous types which show up early in the SCM phase but need evolutionary fine-tuning later. The change of replicator properties in an SCM realisation can be seen in the right column of Fig. 3. Unlike in the MCRS simulation from which it has been initialised, the lengths, and thus also the replication rates, of the replicator types converge, after the transient state.

Compared to MCRS, the SCM system shows a more flexible overall behaviour in time: replicator traits fluctuate at a substantial amplitude, but the distinct replicator types tend to keep their demographic and catalytic properties coordinated through vesicle-level selection (Fig. 3). As the replicator compositions of the vesicles may be completely different and cannot converge due to the lack of recombination/exchange of genetic information among them, the vesicle population maintains a relatively high genetic diversity—and thus a high evolutionary potential—besides a relatively stable and homogeneous distribution of metabolic activities on the vesicle level. The different replicator contents ("genotypes") of the vesicle population are free to explore the sequence and composition space while maintaining ecologically stable communities. When a vesicle assembles a replicator community that is better (in terms of its metabolic efficiency and/or assortment risk) than the rest of the recent system, this new community will spread over time until partially replaced by another vesicle type that is even more successful in the new context. This succession of evolving vesicle types is the cause of the fluctuation in average replicator properties that can be observed in Fig. 3. The same mechanism may be responsible for maintaining a higher number of coexistent replicator types in SCM (Fig. 5) than in MCRS (Fig. 4). This means that the protocellular organisation is dynamically stable and capable of maintaining a high evolutionary potential.

The number of cis-promiscuous replicators is quite high in the SCM; species with 3 catalytic activities on one of their complementary strands may persist for long periods (Fig. 3). The other strand of such pairs is typically a functional parasite or a single-activity strand (in the upper left and lower right block in Fig. 5), but some cis-cis pairs show up, too (see the upper right corner of Fig. 5). Not all the cis- and trans-promiscuous types occurring in Fig. 5 are present in each vesicle, but the successful ones always contain metabolically complete combinations, of course. As hosting multiple enzymatic activities on a single strand requires complex structures to evolve, the cis-promiscuous replicators of the SCM are longer on average (Fig. 3) than the single-activity or trans-promiscuous strands that are more common in the MCRS. The parasite burden of the vesicles is relatively low, making the compartmentalised metabolic system a superior selective environment compared to surface-bound metabolic communities.

## Discussion

The environment in which the first enzymes evolved must have differed from the intracellular environment in which they mainly operate today. Even though "membrane first" scenarios for the origin of life have been repeatedly proposed[89–91], no dynamically feasible mechanism is known by which such systems could have evolved into protocellular structures capable of maintaining and reproducing themselves, together with a chemically coupled metabolic reaction network and genetic material which they would encapsulate[64]. Today it is the RNA world scenario[17] that offers the most plausible—or, at least, imaginable—sequence of evolutionary steps from no life to protocellular life. The advantage of the RNA world scenario is that it suggests a solution to two of the notorious problems of the origin at once:

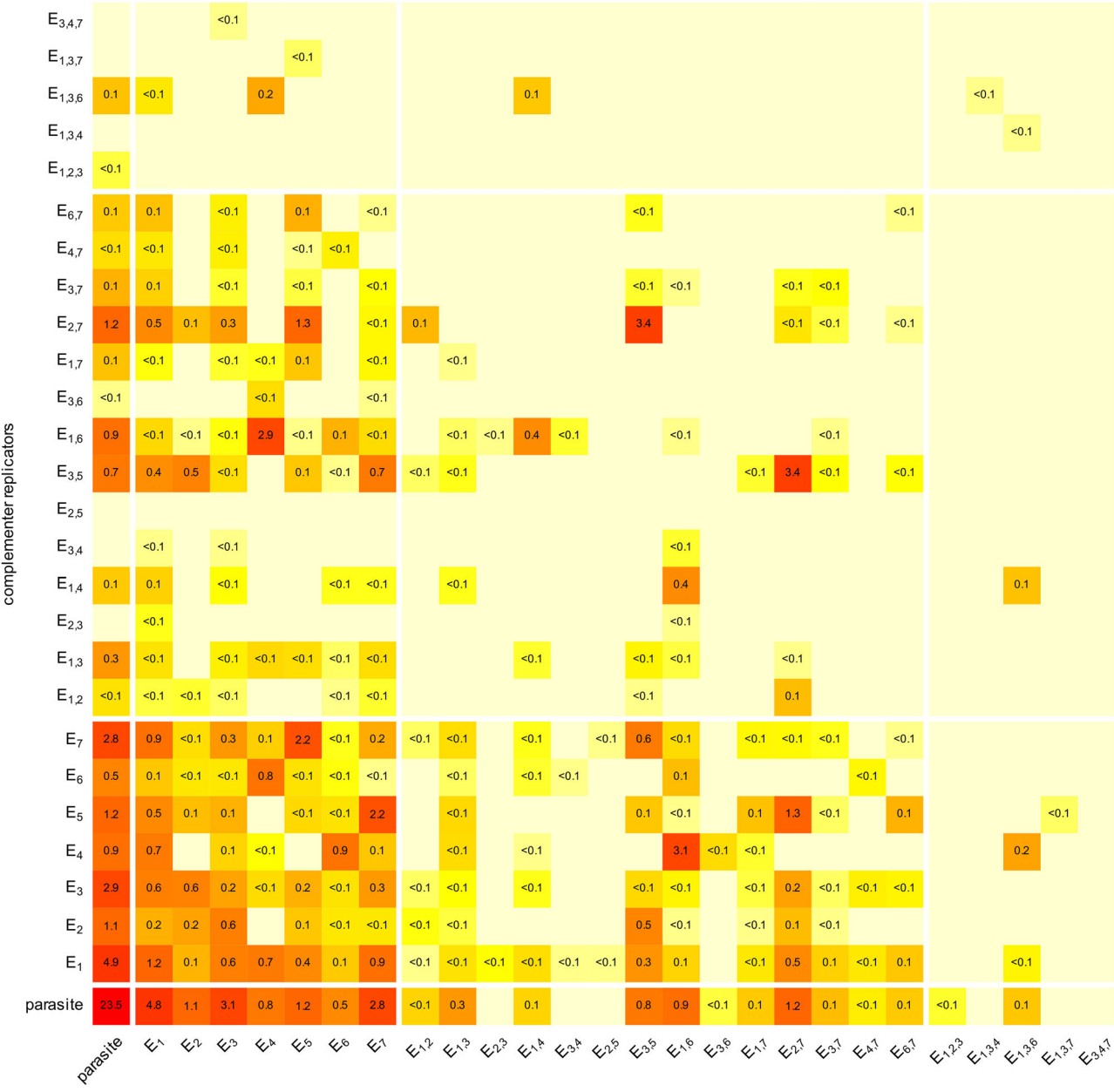

**Fig. 5 | The evolved states of catalytic activities in an SCM simulation after 200,000 generations.** The distribution of replicator types present, categorised by their cis-promiscuity type and the number of their enzymatic activities on one strand (*x*-axis), against the same features of their theoretical (error-free) complementary sequences (*y*-axis) in a model snapshot. Cells represent the trans-promiscuity states of the corresponding complementary pairs of strands. Combinatorically valid cis-promiscuous types absent in the actual set of replicators are not shown (e.g., no $E_{2,4}$ replicator is present in this timeframe of the simulation, so the corresponding row and column are not shown). Colours indicate relative replicator frequencies on a log scale. Numbers inside the cells indicate the percentile frequencies (%) of the corresponding complementary pairs. Real parasites (neither strand is metabolically active) are represented at the bottom left cell. All other cells in the leftmost column and the bottom line represent replicators with only one of their complementary strands active metabolically—the inactive strands of these are functional parasites. Parameters of the simulations as in Fig. 3.

RNA can both catalyse a wide range of metabolic reactions and genetically encode the structures capable of that, i.e., a single RNA molecule can be both an enzyme and the gene of that enzyme[17]. Obviously, the genetic function requires a replicating agent that copies RNA reliably enough to ensure that the information is inherited; in the RNA world context, conveniently such a —thus far undiscovered—universal (and self-replicating) replicase agent should be RNA-based, possibly with cofactors like amino acids or metal ions. Since the many catalytic functions of RNA molecules may also include the production of membrane constituents from small metabolites, it is a credible suggestion that all three main subsystems of cellular life (metabolism, genome, and membrane)[64,73,74,92] could have evolved within dynamically coupled RNA enzyme communities, even if the actual details of the evolutionary process having led there are largely unknown, and possibly will never be known in detail.

Assuming that the catalytic activities of the replicators can withstand the possible weakening effects of the ensuing environmental change[38], the transition between the two dynamically different selection regimes could still compromise the success of the take-off, since a replicator community optimised by evolution in one kind of organisation may be completely unfit in the other. We have shown that it is easier to initialise operational

metabolic replicator communities on a surface than in vesicles and that vesicles with a higher order of complexity can be initialised from replicator sets evolved on the surface. This supports the idea that surface-bound replicator communities could have served as a preadaptation, a stepping stone towards cellular life.

In the MCRS, replicators form discrete and persistent trans-promiscuous classes after the transient stage. In the MCRS phase replicators evolve to achieve the lowest possible degradation rate by keeping their free energy as low as possible. This is supported by the fact that the grid maintains an occupancy of approximately 80% throughout the simulations, which is quite a high value, implying that the main limiting factor for replication is the low frequency of empty sites in the MCRS. On the other hand, in SCM the average free energy of the evolved metabolic replicators is comparable to, or even slightly higher than, that of random replicators in the same size range. Consequently, their replication rates are relatively high, because the high replicabilities arising from loose folding compensate for the disadvantage of longer sequences due to the length constraints of accommodating multiple activities in frequently occurring cis-promiscuous strands, which is, in turn, needed to decrease the assortment load. That is, replicators in the SCM are less strictly selected for low degradation; instead, they are optimised for faster replication: the SCM is "replication-driven", whereas the MCRS is "degradation-driven".

These two systems employ different optimisations to be able to maintain a functioning metabolism and reduce the assortment load. In the MCRS, the effect of different replication rates can be balanced by spatial segregation. Indeed, a considerable portion of the surface area lacks effective metabolism (in the simulation in Fig. 3, at $t = 200,000$, only 48.8% of the grid sites has $M > 0$). In areas where a necessary type (usually that with the lowest replicability) is excluded by others due to the unequal replication rates (competitive exclusion), none of the replicators can replicate. As a result, these areas are soon cleared and available for colonisation from metabolically complete localities, unless the missing enzymatic activity is reintroduced by diffusion. SCM selects more strictly against high variance in replicability, as its replicators have to evolve similar replication rates to maximise the number of functional offspring vesicles, i.e., to minimise the assortment load during vesicle fissions. The assortment load is also responsible for the high level of cis-promiscuity: replicators integrate more (but less efficient) activities (see "Methods") into the same strand, increasing the probability of offspring vesicles having functional metabolisms. That is why the instantaneous variance of replicability ($R$), sequence length and the minimum of Gibbs free energy ($\Delta G_{min}$) is low at any time, and it also explains their coordinated, canonical change (coevolution) over a longer period of time. From an even longer evolutionary perspective, this mechanism could have been replaced by the formation of real chromosomes and a sophisticated gene regulatory mechanism.

Both MCRS and SCM are highly evolvable, but they follow different evolutionary paths. While the MCRS maintains a constant and stable regime in all of its relevant variables after the transient stage, the SCM exhibits a higher evolutionary potential. This is indicated by its higher variation of cis/trans-promiscuous replicator types within an evolved population of vesicles (compare Figs. 4 and 5) and in the rapid fluctuations in average replicator traits (see Fig. 3). Both higher evolvability and cis-promiscuity (as a first step towards chromosomatization later) indicate that the vesicular context is a better candidate for involving/internalising new catalytic functions necessary for the evolution of modern, cellular life. On the other hand, as the successful initialisation of SCM is very improbable with a higher number of mandatory enzymatic activities taken even from a catalytically active, but unevolved, (initial) pool of random sequences, it seems reasonable to suppose that the SCM was booted up from another, already evolved replicator community—possibly from a prebiotic system of surface metabolism, which is more likely to have emerged from a pool of unevolved replicators.

Although it is possible to imagine an RNA replicator community driving a metabolic reaction network that produces monomers for RNA replication[93], as well as components for the membrane compartments encapsulating them, it is difficult to see how such a complex system could be assembled all at once from scratch. We have shown in this model that a relatively small proportion of the replicator communities sampled from an evolved MCRS is viable in the new 3D environment of the SCM, and this scenario is plausible given that the transition had to happen only once in a tremendously long time. It provides a dynamical explanation for the transfer of a surface-born core metabolism into vesicles. Thus, our model may also provide the theoretical basis for three different directions of empirical experiments. First, it assumes a long process of (co)evolution and selection of metabolic ribozymes on mineral surfaces (towards MCRS) (c.f. ref. 38) which calls for an experimental approach. Second, the encapsulation of an already functioning evolved metabolic ribozyme community into a membrane vesicle from the surface could also be experimentally approached e.g., on the basis of Gözen's studies[76]. From an evolutionary point of view, the chemical components of the membrane vesicles should be (by-)products of the metabolic system itself, ensuring the coordinated, simultaneous growth of the replicator community and the vesicles wrapping them. Third, the evolution of ribozymes within vesicles will continue to be an important line of empirical research, with some significant results already achieved[94–98].

## Methods

This section is divided into three main parts. First, we present the details of the metabolic replicator system common to both the MCRS and the SCM: the structural rules defining the functions of the replicators, the algorithms of their replication, degradation and catalytic effect, and the features of the (implicit) metabolism itself. We then define the rules and the parameters of the population dynamics for the two models in separate parts.

### Replicator dynamics and metabolism (common for both MCRS and SCM)

Replicators are represented as RNA molecules of mutable lengths and sequences. Based on their 4-nucleotide-based sequences we calculate their most stable 2D secondary structures corresponding to the folding with the lowest Gibbs free energy ($\Delta G_{min}$) using the ViennaRNA Package[99]. This provides an emergent genotype-phenotype mapping based on physical principles so that the effect of mutations and trade-offs between distinct traits of replicators can be modelled without relying on overly arbitrary assumptions.

Each replicator has the following dynamical traits calculated as functions of length ($L$), minimal free energy ($\Delta G_{min}$) and secondary structure: degradation rate ($P_{deg}$), folding probability ($P_{fold}$), replication rate ($R$) and enzymatic activities ($a_1, \ldots, a_A$) in a predefined (implicit) set of metabolic reactions.

### Degradation and replication

The degradation rate $P_{deg}$ of a given replicator is a sigmoid function of the minimum free (folding) energy $\Delta G_{min}$ of its actual sequence, with minima and maxima of $P_{deg}$ at 0.1 and 0.9 and slope of $d$ (see Eq. 1). It is the probability of a replicator being degraded during a single timestep of length $\Delta t$.

$$P_{deg} = 0.1 + 0.8e^{d\Delta G_{min}} \quad (1)$$

The folding probability $P_{fold}$ or foldedness of a sequence is the proportion of its lifetime spent in its most stable secondary structure, which is calculated using the corresponding Boltzmann statistics. $P_{fold}$ is calculated (Eq. 2) as

$$P_{fold} = 1 - \frac{1}{1 + e^{c\Delta G_{min}}}, \quad (2)$$

where scaling factor $c$ ensures that at $\Delta G_{min} \leq E_{min}$, $P_{fold}$ is 1. $E_{min}$ is the empirical minimum folding energy as determined by a numerical experiment[85]: it is the lowest Gibb's free energy found in a large sample of random sequences of lengths 15 to 75 nucleotides. Note that it is only in this dominant folded state that the replicator is assumed to express enzymatic

activity (if it has any—see the next section), and the same compact folded structure inhibits the strand's replication, imposing a feasible trade-off between catalytic activity and replicability.

The replicability (or the replication rate) of a sequence is based on both its length and its folding probability, as replication can only occur when the replicator is in an unfolded state (which occurs with probability $1 - P_{fold}$, ignoring the folding energy of alternative, less compact secondary structures). The replication time of a template sequence partly depends on its length $L$, linearly. Equation 3 considers the time needed for the initialisation of replication ($b_1$)—which is a length-independent step—and the linear time dependence of nucleotide additions ($b_2$)—which is length-dependent.

$$R = g\frac{l + (1 - P_{fold})}{b_1 + b_2 L} \qquad (3)$$

where $g$ and $l$ are scaling parameters. The sequence of the copy is complementary to that of the template, so getting an identical copy of a replicator requires two consecutive error-free replications. During replication, three types of mutation can occur: substitutions (where the base built into the copy is not the complementary pair of the corresponding base in the template); insertions (a new base is added to the template strand) and deletions (a base of the template is skipped during replication). The per-base probabilities of these mutations are, respectively, $p_{sub}$, $p_{ins}$ and $p_{del}$. Due to the mutations, the secondary structure and the corresponding Gibbs free energy of the copy may change, sometimes drastically, compared to the original, mutation-free sequence. Even a single mutation at the right place can completely spoil the original secondary structure (and, therefore, that of its complementary strand too). Still, most mutations cause only minor changes in the secondary structures of evolved sequences. The change in $\Delta G_{min}$ and sequence length $L$ alters the degradation rate $P_{deg}$, the folding probability $P_{fold}$, and the replication rate $R$ of the replicators.

### Enzymatic activities

RNA macromolecules can catalyse several chemical reactions[19]. So far, no correct algorithm has been proposed to determine enzymatic activities from RNA sequences or secondary/tertiary structures. Thus, we have implemented a series of arbitrary rules to assign (metabolic) enzymatic activities to certain combinations of primary and secondary RNA structural motifs. To have any catalytic activity at all, a replicator has to feature at least one compulsory motif in its dominant secondary structure: a terminal loop or an interloop of length $n$. In the middle of this loop, there are 3 nominal bases which contribute the most to the strength of the given enzymatic activity. Based on how many of these bases comply with the rules (0, 1, 2 or 3), the base activity ($\bar{a}_i$) of motif $i$ is 0, 0.1, 0.8, and 1, respectively. The rest of the bases in the loop motif may also contribute to the net enzymatic activity $\alpha$ depending on their GC content ($n_{GC}$) as a sigmoid function with slope $\xi$:

$$\alpha_i = \bar{a}_i + \frac{1}{2}\frac{1}{1 + e^{n_{GC}\xi}} \qquad (4)$$

Folding probability ($P_{fold}$) also affects enzymatic activity (besides degradation rate and replicability): the more stable a secondary structure is, the more time it can spend catalysing the reaction, so to get the net enzymatic activity ($a_i$) of the combined motif $i$, $\alpha_i$ is multiplied by $P_{fold}$ of the replicator to which the motif belongs (Eq. 5).

Catalytic promiscuity is allowed in the model. A replicator is called trans-promiscuous if it contains at least one motif with enzymatic properties on each of its complementary strands (not necessarily the same on the two). These activities are independent of each other, as they reside on separate entities. On the other hand, cis-promiscuous replicators contain more than one enzymatically active motif on the same strand. In this case, the two concurrent motifs constrain each other functionally, as one enzyme cannot catalyse multiple different reactions at the same time, so these activities are modulated in a sub-additive way due to time-sharing between the two catalytic functions; the parameter $\sigma$ defines the "strength" of sub-additivity.

**Table 2 | The rules applied to determine the strength of a number of different enzymatic activities**

| Number | Motif type | Loop length | Active centres | Symbol |
|---|---|---|---|---|
| 1 | loop | 7 | AGC | ⑦ |
| 2 | interloop | 11 | CCG | ⑪ |
| 3 | interloop | 5 | GCG | ⑤ |
| 4 | interloop | 15 | CCC | ⑮ |
| 5 | loop | 15 | UGU | ⑮ |
| 6 | interloop | 13 | CUG | ⑬ |
| 7 | loop | 9 | CUU | ⑨ |
| 8 | loop | 17 | UAA | ⑰ |
| 9 | interloop | 9 | CUC | ⑨ |
| 10 | loop | 13 | AUG | ⑬ |

Only the first 10 rules are included, as our investigations used only this many. The corresponding symbol is used to indicate the size and type of the loop in figures.

With $m$ concurrent catalytically active motifs on the same strand, we define the $i$-th activity as

$$a_i = P_{fold}\frac{\alpha_i}{m^\sigma} \qquad (5)$$

The (arbitrary) rules determining enzymatic activities were randomly generated (and used throughout all subsequent simulations) by the following algorithm: (1) loops and interloops of random sizes were drawn from the [5, 17] length interval (only odd numbers), (2) random base triplets were assembled by drawing three bases each with equal probability. The motifs and activities that we used in the simulations are listed in Table 2.

Enzymatic activities are more prone to mutations than degradation or replication rates. Even a single base insertion or deletion in the catalytic motif that barely changes the secondary structure or $\Delta G_{min}$ may turn on or off an enzymatic activity entirely. On the other hand, base substitutions which do not affect the secondary structure might also fine-tune the strength of enzymatic activities due to the redundancy of active centre bases or through the optimisation in the GC base content of the catalytic loop. This makes the gradual evolution of enzymatic activities feasible.

### Metabolism

The core assumption of the models applied in this study is that RNA replicators catalyse the reactions of a common metabolism ($M$) that supplies the set of monomers they need for their own replication (see "Introduction", Fig. 1). This common metabolism is a hypothetical (and, for now, implicit) reaction network, each enzymatically aided reaction step of which is supposed to be essential in the production of the building blocks (monomers) of the replicators. If any one of these reactions lacks its specific enzyme, then the metabolic network fails to produce monomers, which in turn arrests replication in the given local context. This functional requirement is implemented in the models by applying a metabolic function of replicator $i$ ($M_i$) that is calculated as the geometric mean of the sum of all enzymatic activities:

$$M_i = \sqrt[A]{\prod_{e=1}^{A}\sum_{r \in N_{met}} a_{r,e}} \qquad (6)$$

where $a_{r,e}$ is the enzymatic activity $e$ of replicator $r$ belonging to the same *metabolic environment* (see later) that contains $N_{met}$ replicators. $A$ is the number of the different compulsory ribozyme activities in a metabolically complete environment (system size). Replicator $i$ has a claim $C_i$ for being replicated, which is proportional to its metabolic support $M_i$ and its replicability $R_i$ (Eq. 3):

$$C_i = R_i M_i \qquad (7)$$

The value of the metabolic function might vary to a great extent depending on the spatial context of the replicator; this will be discussed later.

### Population dynamics of the Metabolically Coupled Replicator System (MCRS)

RNA replicators and organic metabolites may adhere to mineral surfaces[31,33,100], which would result in limited surface diffusion and, consequently, spatial constraints on the chemical interactions among such surface-bound substances. The probability of a pair of macromolecules to interact depends on their spatial proximity if their mobility on the surface is constrained. The MCRS implements this form of two-dimensional spatial segregation by defining the surface as a square lattice of cellular automata, with each cell of the $Z \times Z$ lattice either empty or occupied by exactly one replicator. Assuming toroidal topology for the lattice (with opposite margins fused) avoids edge effects at the price of introducing spatial periodicity. Both replicators and

metabolites have a limited spatial range of interaction on the lattice. Replicators can place a complementary copy of their sequence into empty sites in their replication neighbourhood (an area of fixed size and shape centred on the focal replicator, containing $N_{rep}$ cells), and their enzymatic activities have an effect only within the metabolic neighbourhood ($N_{met}$) of the replicator. These two neighbourhoods ($N_{rep}$ and $N_{met}$) may be of different sizes, since the replication neighbourhood depends on the surface mobility and desorption of the macromolecular replicators, whereas the metabolic neighbourhood is determined by the surface diffusion rate and the desorption rate of the metabolites[39,80,86,88]. For our neighbourhood size and shape definitions, see Supplementary Note 2.

### Replication in the MCRS

If an empty site is chosen for update, the replicators in its replication neighbourhood ($N_{rep}$) compete for the opportunity to place a complementary copy of themselves to the empty site. For each of the applicants their replication claims $C_i$ are calculated (Eq. 7). The empty site also has a fixed claim $C_e$ to remain empty. The broken-stick method is used to determine which (if any) of the copies of the competing replicators occupies the site. The probability that replicator $i$ will replicate to the empty site is:

$$P(i) = \frac{C_i}{C_e + \sum_{r=1}^{N_{rep}} C_r}, \qquad (8)$$

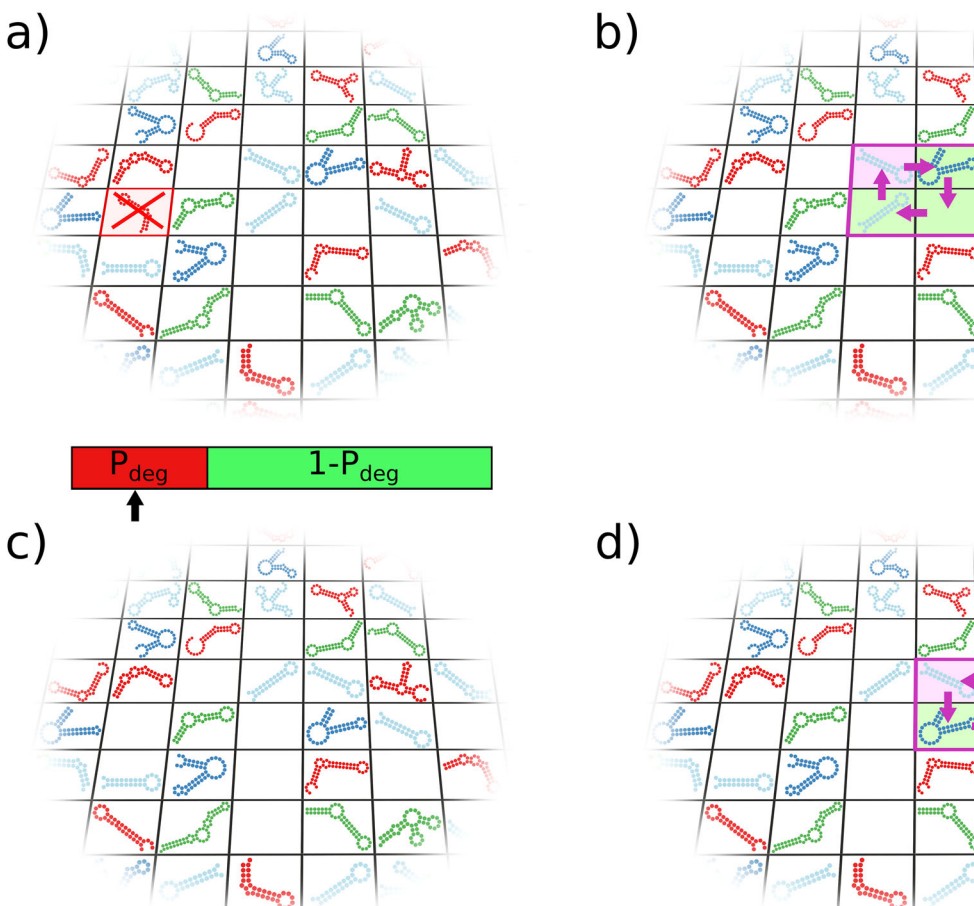

**Fig. 6 | Degradation and replicator diffusion in the MCRS. a** If an occupied site is chosen for update (crossed cell with red border), it degrades with probability $P_{deg}$. The bar below represents the broken-stick method we applied to decide if degradation happens (red area) or not (green area) and the arrow represents the decisive random number. **b** During a diffusion step, a grid site (focal site with purple background) and its Toffoli-Margolus diffusion neighbourhood (green background) are chosen. Then, the contents of the four grid sites in the diffusion neighbourhood are rotated in either the clockwise (**b, c**) or the anti-clockwise direction (**d**).

where $\sum_{r=1}^{N_{rep}} C_r$ is the sum of claims of all the replicators present in the replication neighbourhood. The probability of no replication is

$$P(\text{empty}) = \frac{C_e}{C_e + \sum_{r=1}^{N_{rep}} C_r}. \qquad (9)$$

If a replicator is drawn for replication its copy will adhere to the empty site.

The space-related parameters specific to the MCRS model and their default values are listed in Table 1.

### Surface movement of the replicators

The surface movement of the replicators is modelled by $D \times Z \times Z$ Toffoli-Margolus update steps[101] per generation, where $D$ is the number of diffusion update steps applied after each degradation/replication update step. In a diffusion step, we choose a random Toffoli-Margolus neighbourhood (a cluster of four adjacent cells forming a square with the randomly chosen cell in its upper left corner) and rotate it in the clockwise or the anti-clockwise direction with 50–50% probability (Fig. 6).

### Population dynamics on the surface

During one epoch (or generation) of the simulation, we asynchronously update $Z \times Z$ randomly chosen cells, so that every cell of the lattice is updated once per generation on average. If the updated cell is occupied by a replicator, then that replicator degrades with probability $P_{deg}$ (Eq. 1, Fig. 6), leaving the site empty. If the updated cell is empty, then a complementary copy of one of the replicators from $N_{rep}$ of the focal empty site may occupy it (Eq. 8, Fig. 7).

### Initialisation of the MCRS simulations

To initialise the simulations, a large pool of random sequences (lengths are drawn from a Poisson distribution with $\lambda = 45$) was generated. In order to ensure that the system takes off, these pools were "enriched": the replicators with at least one enzymatic activity were multiplied until 80% of the replicators were catalytically active. This is a technical necessity due to the finite, relatively small lattice size we had to use for computational reasons: obviously, there has to be at least one metabolically complete "seed" metabolic neighbourhood on the lattice for the system to be viable. On an infinite lattice, this would happen with probability 1; thus, the "enrichment" procedure would be superfluous. The simulations started with the lattice filled up to 80% with sequences randomly sampled from the enriched pool.

### Population dynamics of the Stochastic Corrector Model (SCM)

We have implemented MCRS-like metabolism and dynamics in the SCM context[50,78]. $n_V$ replicators are encapsulated in $N$ compartments, with each individual compartment containing 0 to $S - 1$ replicators at the end of an iteration step, regardless of their length or type. These replicators

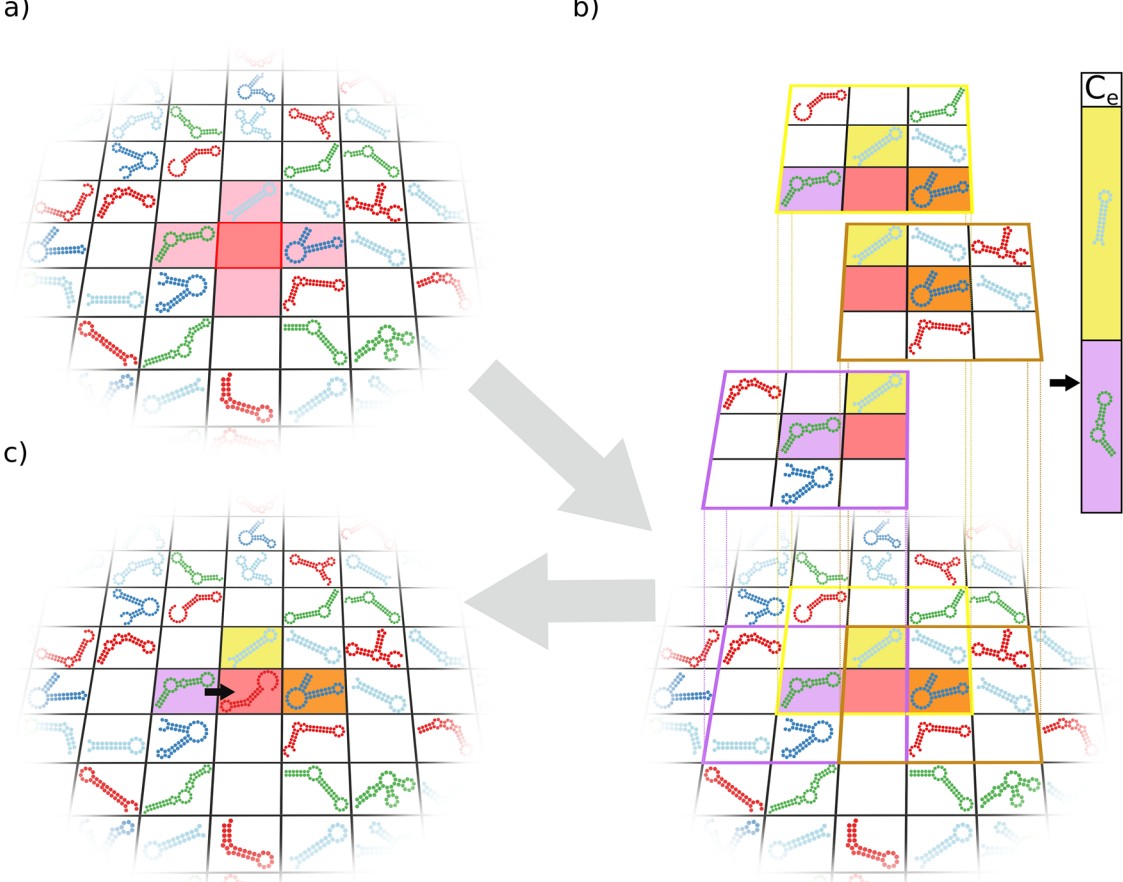

**Fig. 7 | MCRS replication. a** During an update step, an empty site is randomly chosen (red background with red border). Its neighbours in its replication neighbourhood (light red background) may claim to replicate to the empty site (in this example, only 3 replicators are present in the 4 sites of the replication neighbourhood). **b** Each replicator in the empty site's replication neighbourhood may use the catalytic help of other replicators belonging to their own metabolic neighbourhoods (floating 3 × 3 grids with yellow, orange and purple borders). Note that these metabolic neighbourhoods overlap! The claim $C_i$ of replicator $i$ depends on its individual metabolic support ($M_i$) and replication rate ($R_i$). The metabolic neighbourhood of the "orange" replicator is incomplete, therefore its claim is 0. The colour bar on the right represents the broken stick, with pieces for the claims of the two remaining contestants and the claim of the empty site to remain empty. The arrow shows the actual decision on the winner (the "purple" replicator in this example). **c** Based on the decision in (**b**), the "purple" replicator puts a complementary copy of itself into the empty site.

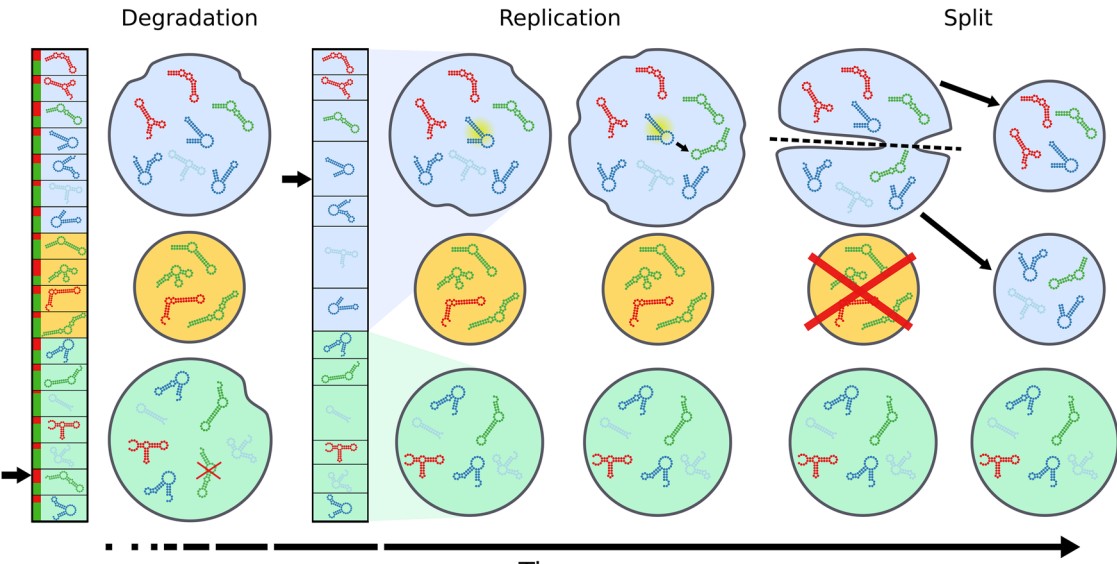

**Fig. 8 | The dynamics of the SCM.** In the degradation step, a degradation (red) or a survival (dark green) event is randomly chosen from the pooled $\{P_{\text{deg}}, (1-P_{\text{deg}})\}$ values of all replicators (left column), independently of which vesicle the corresponding replicator belongs to. (The black arrow indicates the random number drawn in the broken stick algorithm; in this specific case, a replicator in the green vesicle was chosen for degradation). Replication is context-dependent, so individual replication propensities ($C_{i,v}$) are the product of the replicators' replication rates ($R_{i,v}$) and the metabolic efficiencies in the encapsulating vesicles ($M_v$). If, after a replication step, a vesicle's content reaches the split size ($S$), it splits hypergeometrically, and one of the daughter vesicles overwrites another one chosen at random (here: the yellow one).

drive the common metabolism $M$ which will contribute to the replication of the replicators inside the compartment in a way exactly corresponding to the algorithm applied in the MCRS (see Eq. 6), in the metabolic context of a vesicle. For the parameters and variables of the model see Table 1.

During the stochastic simulation of the dynamics of replicators, each update step consists of two consecutive reactions involving broken-stick decisions: a degradation and a replication step. Each replication step may be followed by a vesicle split event if it is required.

1. Degradation:
   In each update step, a random replicator is chosen to degrade. This can be interpreted as a broken-stick step in which all replicators participate (regardless of the vesicle to which they belong) with two propensities: $P_{\text{deg}}$ for degradation and $1-P_{\text{deg}}$ for staying alive (Fig. 8).

2. Replication:
   All replicators within their own vesicle $v$ express their respective claims ($C_{i,v}$) to replicate according to their own replication rates ($R_{i,v}$) and the metabolic support provided by the ribozymes in their own encapsulating vesicles ($M_v$).

$$C_{i,v} = M_v R_{i,v} \tag{10}$$

During an update step, all claims are pooled, and one replicator is chosen to be copied using the broken stick algorithm (Fig. 8): the probability of replicator $i$ in vesicle $v$ to replicate during an update step is

$$P(i,v) = \frac{C_{i,v}}{\sum_{V=1}^{N} \sum_{r=1}^{n_v} C_{r,V}}. \tag{11}$$

If a vesicle lacks at least one of the necessary enzymatic activities, then $M_v = 0$, therefore the claims of all replicators in vesicle $v$ will be 0 and no replication can occur.

3. Splitting:
   If $n_v = S$ after a replication step, the compartment will split into two daughter compartments (Fig. 8). During a splitting event, the replicators of the parent vesicle are distributed among the daughter

compartments in a hypergeometrical manner. A randomly chosen offspring vesicle will take the place of the original compartment, while the other will overwrite a randomly chosen compartment (which may even be its own sister vesicle by chance). This guarantees that the number of compartments ($N$) remains constant during the simulation.

We define a timestep as $\frac{N(S-1)}{2}$ consecutive update steps for the SCM timescale to be comparable to that of MCRS. In MCRS a timestep consists of $Z \times Z$ update steps, so the number of elementary events equals the maximum number of replicators there. In SCM the maximum number of replicators is $N(S-1)$, but an update step contains two events (both a degradation and a replication). Several other time scaling algorithms were also tried for the SCM, with no substantial differences in their outcomes.

## Transition between SCM and MCRS

The "melting" vesicle formation mechanism (see "Introduction") was implemented by choosing random circular areas on the lattice of an evolved MCRS simulation and enclosing their replicator content into separate vesicles. The diameter of the sampling circles was chosen from a unimodal Gaussian distribution.

We have initialised the SCM with compartments in 4 different ways:

- Random initialisation: filling up each of $N$ compartment with $n_{\text{init}}$ random replicators from an enriched pool (as done in MCRS initialisation).
- Asynchronous transition from one vesicle: a single compartment was filled with a sample from a steady-state MCRS simulation and loaded into SCM, along with $N-1$ empty compartments, as the initial state of the simulation.
- Simultaneous transition from multiple vesicles: $n_{\text{sampling}}$ compartments were sampled from a steady MCRS simulation and loaded into SCM, along with $N-n_{\text{sampling}}$ empty compartments, as the initial state of the simulation.
- Sequential transition: within a time interval of length $\Delta t_{\text{trans}}$, $n_{\text{sampling}}$ compartments were sampled in each generation from the lattice of a

running MCRS simulation that was synchronised with the running SCM simulation. These compartments have replaced the (initially empty) compartments of the SCM.

## Reporting summary

Further information on research design is available in the Nature Portfolio Reporting Summary linked to this article.

## Data availability

The results of Fig. 3 are included in our GitHub repository[102]: https://github.com/danithered/mcrscm.

## Code availability

To make our results reproducible, all C++ source codes with usage instructions are included in our GitHub repository[102]: https://github.com/danithered/mcrscm.

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

## Acknowledgements
The project was supported by the ERC Synergy MiniLife Project (Grant No. 101118938) and the HUN-REN SA-59/2021 grant. We are grateful for the possibility of using HUN-REN Cloud[103] (https://science-cloud.hu/), which helped us achieve the results published in this paper. The authors would like to thank Géza Meszéna and Eörs Szathmáry for helpful discussions.

## Author contributions
Conceptualisation: D.V., T.C., A.S., B.K. Methodology: D.V., T.C., A.S., B.K. Investigation: D.V. Visualisation: D.V. Funding acquisition: A.S. Project administration: A.S. Supervision: T.C., A.S., B.K. Writing—original draft: D.V., T.C., A.S., B.K. Writing—review and editing: D.V., T.C., A.S., B.K.

## Funding

## Competing interests
The authors declare no competing interests.
