## [Transparent Peer Review file · Communications Biology]

The dynamics of prebiotic take-off: the transfer of functional RNA communities from mineral surfaces to vesicles

Corresponding Author: Dr Tamás Czárán

Version 0:

Reviewer comments:

Reviewer #1

(Remarks to the Author)

Vörös and colleagues present a theoretical analysis of two models of prebiotic evolution, MCRS and SCM, in the context of a transition from RNA replicator/ribozyme communities existing in 2D, specifically, on mineral surfaces, to protocells. They conclude that MCRS is a better model for the earliest stages of evolution, presumably, random RNA ensembles on mineral surface, whereas SCM works better at the protocellular stage, and that the transition between the two models is feasible.

I do not have immediate problems with the modeling part, but I am just not convinced by the premise. Is the mineral surface stage feasible AND necessary? The authors themselves note that mineral surfaces would not provide particularly efficient compartmentalization. They also correctly note that vesicles can spontaneously form under abiogenic conditions and can accumulate various compounds including nucleotides as shown by Szostak and others. I think it is simply far more plausible that pre-biotic evolution started with such vesicles rather than with mineral surfaces. I am afraid that the use of this contrived premise, mineral surfaces as the initial milieu of prebiological evolution, greatly diminishes the interest of this analysis of models which in itself is technically valid.

The manuscript is well written but is very verbose, especially, the Introduction

Reviewer #2

(Remarks to the Author)

The manuscript entitled "The dynamics of prebiotic take-off: from mineral surfaces to vesicles" by Voros et al. submitted for publication to Nature Communications Biology uses a computational approach to address questions about the origin of the first protocellular form of life. The premise invokes the RNA World Model by an explicit dynamical interface that simulates the transition of a metabolically cooperating RNA replicator community from a mineral surface into a population of membrane vesicles. The two agent-based models: the Metabolically Coupled Replicator System (MCRS) and the Stochastic Corrector Model (SCM), are built on principles of systems chemistry, molecular biology, ecology and evolutionary biology.

Questions about the origin of the first protocellular forms of life are very important in research on the Origin of Life (OoL) on earth. As the authors state, "The origin of life is usually approached either from a theoretical perspective by examining the rules of self-organization, or by focusing on the synthesis of biomolecules in a more empirical setting. An ideal, and certainly more fruitful approach would be one in which theoretical studies would instruct empirical work, and experiments would channel theory". This is very important and a key to the significance and importance of theoretical and computational approaches instructing empirical work. While emphasizing in outline this important transition, the authors do not however discuss nor present information about how their work might provide insight into empirical approaches. They do a reasonable job of referring to much important literature on prebiotic and abiotic chemistry leading to protocellular life but do not integrate with possible insights to move things forward from their studies. A few additional important papers on the prebiotic and abiotic origins of protocells might have been referenced, e.g. O'Flaherty D, Kamat NP, Mirza FN, Li L, Prywes N, and Szostak JW. Copying of mixed sequence RNA templates inside model protocells. *J. Am. Chem. Soc.*, 2018; 140:5171-5178. As a further example, from P. 3 of the manuscript: "Could the transition between surface and cellular environments be even possible? The prebiotic synthesis of vesicle building blocks (e.g. lipids) is feasible, as is their spontaneous organisation into vesicles both in water and on surfaces. There are hypothetical encapsulation mechanisms, some with experimental support, such as the wetting-dewetting cycles of vesicles or the wet-dry/freeze-thaw cycles, which can be assumed to support the transfer of surface-bound replicator communities into vesicles. This environmental change does not affect the enzymatic properties of the ribozymes, but it is still debatable whether the transferred communities could survive the 2D-3D transition, changing selective regimes in terms of community topology, functional diversity, stability and enzymatic properties. The present study aims to investigate the hurdles of such a transition by focusing on the corresponding changes in the dynamics

and the organization of replicator communities. The 'hurdles' that the present study "aims to investigate" could have been presented in more detail with thoughts about ways to transition from theoretical analysis to empirical settings which the authors themselves describe as a "more fruitful approach".

Overall the methodology and concept of the manuscript seem reasonable and the methods are sound and thorough. Availability of code for those who wish to reproduce simulations is useful. There is an increasing need for more theoretical-driven and computational approaches that capture what the first populations of RNA replicators might have been like. Combining both methods and showing the versatility of replicator types that emerge in the different conditions is beneficial. That it addresses core issues in the origin of life, i.e., the origins of cellularity, catalytic RNA, RNA self-replication is important and useful. While useful, the paper and conclusions present progress on the authors' previous work and in that sense is not truly original. The two models implemented in the paper are not new, and the authors themselves cite their repeated development and uses of them. It would therefore be worth explicitly stating the novel aspects of the study with respect to implementing the methods, i.e., in combining the models to make more theoretical insights on evolutionary stages during the origin of life. One item where useful value is added is the interesting proposition that the early evolution of self-replicating molecules could have resembled the transition from MCRS-like to SCM-like replication, and how the simulations conducted demonstrate the feasibility of this option. It would be useful for the authors to clarify whether these models have been used for this specific purpose before, and thus whether their conclusions are novel.

Another thing that could be addressed in more depth would be to mention other RNA replication computational/simulation models in the literature and to better contextualize the key added value(s) of combining both models. Virtually all of the references to the two models discussed are co-authored by the authors and their collaborators. What simulations from others have been done on these questions? As well as additional reference to the work of Szostak on protocellular evolution mentioned above, there is also no mention of other key contributors to questions of the OoL, e.g. Cech, Altman, Koonin, etc. Eigen is briefly mentioned but it would be good to elaborate more on his specific work in relation to OoL simulations. A more comprehensive review of the literature and situating the model and findings in the context of the wider field would be appropriate and useful. Because the computational aspects of the paper are difficult for the non-expert to follow, as it stands now, this paper is of value mostly to researchers with computational expertise. To be more useful to a broader audience with interest in questions of the Origin of Life, better transitioning between theoretical approaches and conclusions to expand on the literature and possible future approaches might be appropriate and valuable.

Reviewer #3

(Remarks to the Author)

Version 1:

Reviewer comments:

Reviewer #1

(Remarks to the Author)

I remain more skeptical than the authors of this paper regarding the role of mineral surfaces in the origin of cells, but the authors are entitled to their view, and the revision is acceptable.

Reviewer #2

(Remarks to the Author)

Comments to Authors

These comments reflect the response of this reviewer to the rebuttal by the authors to the initial review of the manuscript "The dynamics of prebiotic take-off: from mineral surfaces to vesicles" by Dr Czárán and colleagues, which has been revised for Communications Biology in response to previous comments. The authors present a theoretical analysis of two models of prebiotic evolution, MCRS and SCM, in the context of a transition from RNA replicator/ribozyme communities existing in 2D, specifically, on mineral surfaces, to protocells. They conclude that MCRS is a better model for the earliest stages of evolution, presumably, random RNA ensembles on mineral surface, whereas SCM works better at the protocellular stage, and that the transition between the two models is feasible. In their rebuttal, the authors present what we see as a minimal response to the major concerns of Reviewer 1.

The rebuttal focuses on a mineral surface and little else. The authors did much else so this essentially exclusive focus on the issue of whether a mineral surface is 'feasible and necessary' may still be overstated. It is a model so it may not be the most relevant if it is 'necessary'. The rebuttal response still appears to be overstated. A better response may be of the form 'it is not absolutely necessary but it provides a useful theoretical framework for thinking about the question. A theory is to inform thinking, understanding and future questions, not to determine what 'must' have occurred.

The authors correctly respond that a joint mechanism of compartmentalization and catalysis is useful, since parallel origins of the two processes is less plausible. They then however conclude that a mineral surface *must* be the solution. There are clearly other possible models, some already proposed and others under intensive present investigation that can occur. A better way to frame this might be that the mineral surface provides one possible solution but it is not *the* solution. Regarding comments of Reviewer 2, the authors did not seem to sufficiently address the first paragraph in the review. They

mention adding three lines to address the study by Gozen that was “to a large extent, our motivation for this theoretical approach”. They have addressed this now but more than a 3-sentence response would be justified for the ~2 paragraphs of feedback provided by Reviewer 2.

The authors note that the ‘real innovation’ was “the study of the transitory dynamics that is feasible because the two different scenarios are algorithmically possible to link.” This is now partially clarified, but it is not clear that the models implemented were already in existence. This has now been highlighted in lines 169-181 which is useful. In this context, the work presented in the manuscript is not unique because the models implemented were already in existence by the authors themselves and by others. Nonetheless, this work does advance knowledge about this question and publication is warranted.

In thinking about the ‘necessity’ of mineral surfaces, this model can be useful for thinking about the origin of life even if mineral surfaces were not the mode of pre-membrane compartmentalization. The authors can still make a case for their model without overstating its ‘necessity’.

It seems a major motivation of the study is to explore the dynamics of the ‘2D to 3D’ replication surface, i.e., replication on a ‘flat’ mineral surface evolving into a 3D space in which replication occurs. This appears more strongly emphasized in the present manuscript version provided compared to the initial submission. But due to the manuscript's overly verbose style and somewhat lack of a coherent and clear message, the writing needs additional strong editing.

The authors have added some of the references suggested by Reviewer 2 but are still missing some key references, such as Cech and Altman’s early 1980s papers, and more recent work from Koonin.

The tracked/edited manuscript provided to the reviewers is very difficult to follow and needs extensive editing by the journal copy editors which is beyond the scope of this reviewer. The grammar and writing style require significant editorial improvement. As well as standard editorial improvement, a clear, strong, and supportable conclusion of the work would help. This reviewer suggests that the scholarly merit of the work could warrant publication provided the writing style and clarity of the presentation is significantly improved in consultation with the Journal copy editors.

Reviewer #3

(Remarks to the Author)

[this reviewer conducted a co-review alongside another reviewer, as such, comments may be duplicated]

These comments reflect the response of this reviewer to the rebuttal by the authors to the initial review of the manuscript “The dynamics of prebiotic take-off: from mineral surfaces to vesicles” by Dr Czárán and colleagues, which has been revised for *Communications Biology* in response to previous comments. The authors present a theoretical analysis of two models of prebiotic evolution, MCRS and SCM, in the context of a transition from RNA replicator/ribozyme communities existing in 2D, specifically, on mineral surfaces, to protocells. They conclude that MCRS is a better model for the earliest stages of evolution, presumably, random RNA ensembles on mineral surface, whereas SCM works better at the protocellular stage, and that the transition between the two models is feasible. In their rebuttal, the authors present what we see as a minimal response to the major concerns of Reviewer 1.

The rebuttal focuses on a mineral surface and little else. The authors did much else so this essentially exclusive focus on the issue of whether a mineral surface is ‘feasible and necessary’ may still be overstated. It is a model so it may not be the most relevant if it is ‘necessary’. The rebuttal response still appears to be overstated. A better response may be of the form ‘it is not absolutely necessary but it provides a useful theoretical framework for thinking about the question. A theory is to inform thinking, understanding and future questions, not to determine what ‘must’ have occurred.

The authors correctly respond that a joint mechanism of compartmentalization and catalysis is useful, since parallel origins of the two processes is less plausible. They then however conclude that a mineral surface *must* be the solution. There are clearly other possible models, some already proposed and others under intensive present investigation that can occur. A better way to frame this might be that the mineral surface provides one possible solution but it is not *the* solution. Regarding comments of Reviewer 2, the authors did not seem to sufficiently address the first paragraph in the review. They mention adding three lines to address the study by Gozen that was “to a large extent, our motivation for this theoretical approach”. They have addressed this now but more than a 3-sentence response would be justified for the ~2 paragraphs of feedback provided by Reviewer 2.

The authors note that the ‘real innovation’ was “the study of the transitory dynamics that is feasible because the two different scenarios are algorithmically possible to link.” This is now partially clarified, but it is not clear that the models implemented were already in existence. This has now been highlighted in lines 169-181 which is useful. In this context, the work presented in the manuscript is not unique because the models implemented were already in existence by the authors themselves and by others. Nonetheless, this work does advance knowledge about this question and publication is warranted.

In thinking about the ‘necessity’ of mineral surfaces, this model can be useful for thinking about the origin of life even if mineral surfaces were not the mode of pre-membrane compartmentalization. The authors can still make a case for their model without overstating its ‘necessity’.

It seems a major motivation of the study is to explore the dynamics of the ‘2D to 3D’ replication surface, i.e., replication on a ‘flat’ mineral surface evolving into a 3D space in which replication occurs. This appears more strongly emphasized in the present manuscript version provided compared to the initial submission. But due to the manuscript's overly verbose style and somewhat lack of a coherent and clear message, the writing needs additional strong editing.

The authors have added some of the references suggested by Reviewer 2 but are still missing some key references, such as Cech and Altman’s early 1980s papers, and more recent work from Koonin.

The tracked/edited manuscript provided to the reviewers is very difficult to follow and needs extensive editing by the journal copy editors which is beyond the scope of this reviewer. The grammar and writing style require significant editorial improvement. As well as standard editorial improvement, a clear, strong, and supportable conclusion of the work would help. This reviewer suggests that the scholarly merit of the work could warrant publication provided the writing style and clarity of the presentation is significantly improved in consultation with the Journal copy editors.

Response to reviewers

Please find our replies to the reviews below *in italics*.

Reviewers' comments:

Reviewer #1 (Remarks to the Author):

Vörös and colleagues present a theoretical analysis of two models of prebiotic evolution, MCRS and SCM, in the context of a transition from RNA replicator/ribozyme communities existing in 2D, specifically, on mineral surfaces, to protocells. They conclude that MCRS is a better model for the earliest stages of evolution, presumably, random RNA ensembles on mineral surface, whereas SCM works better at the protocellular stage, and that the transition between the two models is feasible.

I do not have immediate problems with the modeling part, but I am just not convinced by the premise. Is the mineral surface stage feasible AND necessary? The authors themselves note that mineral surfaces would not provide particularly efficient compartmentalization. They also correctly note that lipid vesicles can spontaneously form under abiogenic conditions and can accumulate various compounds including nucleotides as shown by Szostak and others. I think it is simply far more plausible that prebiotic evolution started with such vesicles rather than with mineral surfaces. I am afraid that the use of this contrived premise, mineral surfaces as the initial milieu of prebiological evolution, greatly diminishes the interest of this analysis of models which in itself is technically valid.

Yes, we are convinced that the mineral surface stage is both feasible and necessary. Its chemical feasibility has been emphasised in theory and demonstrated experimentally by many authors whose work we have cited in the manuscript. Regarding its necessity, we have inserted a few sentences (lines 100-118 in MA with track changes) to explain the two main reasons why the mineral surface phase cannot be omitted from the prebiotic scenario. In essence, one reason is ecological: the metabolically cooperating ribozyme community has to persist (coexist) long enough to have sufficient time to evolve ribozymes producing the building blocks of vesicles. Coexistence requires some form of group structure in the community, and the simplest such structure is that on a surface. The other would be compartmentation in vesicles, but for that to work, one needs a ribozyme set evolved to produce membrane constituents at exactly the rate of the growth of the metabolic ribozyme community. This is the other, evolutionary reason why the surface stage cannot be omitted: in a 3D bulk phase, no group structure can be maintained without compartments. Therefore, no sustained coexistence can be expected there. It is also not sufficient to put together lipids and nucleic acids to get a viable protocell: metabolism and compartmentation must be coordinated, which can occur by gradual adaptation. It is on the surface where there would be enough time for evolving membrane-producing replicators from the parasitic RNA sequences that remain present in the surface model (MCRS) but are kept at bay by surface dynamics.

The manuscript is well written but is very verbose, especially, the Introduction

Thanks, we agree. We have deleted a few superfluous sentences and rewritten many others in the Introduction. We hope the text is more concise and clear now.

Reviewer #2 & #3 (Remarks to the Author):

The manuscript entitled “The dynamics of prebiotic take-off: from mineral surfaces to vesicles” by Voros et al. submitted for publication to Nature Communications Biology uses a computational approach to address questions about the origin of the first protocellular form of life. The premise invokes the RNA World Model by an explicit dynamical interface that simulates the transition of a metabolically cooperating RNA replicator community from a mineral surface into a population of membrane vesicles. The two agent-based models: the Metabolically Coupled Replicator System (MCRS) and the Stochastic Corrector Model (SCM), are built on principles of systems chemistry, molecular biology, ecology and evolutionary biology.

Questions about the origin of the first protocellular forms of life are very important in research on the Origin of Life (OoL) on earth. As the authors state, “The origin of life is usually approached either from a theoretical perspective by examining the rules of self-organization, or by focusing on the synthesis of biomolecules in a more empirical setting. An ideal, and certainly more fruitful approach would be one in which theoretical studies would instruct empirical work, and experiments would channel theory”. This is very important and a key to the significance and importance of theoretical and computational approaches instructing empirical work. While emphasizing in outline this important transition, the authors do not however discuss nor present information about how their work might provide insight into empirical approaches. They do a reasonable job of referring to much important literature on prebiotic and abiotic chemistry leading to protocellular life but do not integrate with possible insights to move things forward from their studies. A few additional important papers on the prebiotic and abiotic origins of protocells might have been referenced, e.g. O’Flaherty D, Kamat NP, Mirza FN, Li L, Prywes N, and Szostak JW. Copying of mixed sequence RNA templates inside model protocells. *J. Am. Chem. Soc.*, 2018; 140:5171-5178. As a further example, from P. 3 of the manuscript: “Could the transition

between surface and cellular environments be even possible? The prebiotic synthesis of vesicle building blocks (e.g. lipids) is feasible, as is their spontaneous organisation into vesicles both in water and on surfaces. There are hypothetical encapsulation mechanisms, some with experimental support, such as the wetting-dewetting cycles of vesicles or the wet-dry/freezing-thaw cycles, which can be assumed to support the transfer of surface-bound replicator communities into vesicles. This environmental change does not affect the enzymatic properties of the ribozymes, but it is still debatable whether the transferred metabolic communities could survive the 2D-3D transition, changing selective regimes in terms of community topology, functional diversity, stability and enzymatic properties. The present study aims to investigate the hurdles of such a transition by focusing on the corresponding changes in the dynamics and the organization of replicator communities. The ‘hurdles’ that the present study “aims to investigate” could have been presented in more detail with thoughts about ways to transition from theoretical analysis to empirical settings which the authors themselves describe as a “more fruitful approach”.

One specific empirical study on exactly the prebiotic take-off has already been accomplished (Gözen, 2021- ref. #49 in the MS) – in fact, it was, to a large extent, our motivation for this theoretical approach. We agree that this should be more emphatic in the Introduction, so we have inserted a short section explaining and citing this work at lines 134-137 in MA with track changes.

Overall the methodology and concept of the manuscript seem reasonable and the methods are sound and thorough. Availability of code for those who wish to reproduce simulations is useful. There is an increasing need for more theoretical-driven and computational approaches that capture what the first populations of RNA replicators might have been like. Combining both methods and showing the versatility of replicator types that emerge in the different conditions is beneficial. That it addresses core issues in the origin of life, i.e., the origins of cellularity, catalytic RNA, RNA self-replication is important and useful. While useful, the paper and conclusions present progress on the authors’ previous work and in that sense is not truly original. The two models implemented in the paper are not new, and the authors themselves cite their repeated development and uses of them.

It would therefore be worth explicitly stating the novel aspects of the study with respect to implementing the methods, i.e., in combining the models to make more theoretical insights on evolutionary stages during the origin of life. One item where useful value is added is the interesting proposition that the early evolution of self-replicating molecules could have resembled the transition from MCRS-like to SCM-like replication, and how the simulations conducted demonstrate the feasibility of this option. It would be useful for the authors to clarify whether these models have been used for this specific purpose before, and thus whether their conclusions are novel.

The two models are indeed essentially the same as those published earlier, except that now we have tailored them to be consistently compatible so that we can compare their dynamical outcomes before, during and after the transition. This includes a substantial change in the SCM so that it handles mortality events on the vesicle level, which was not the case in earlier SCM versions. Now, the two models are comparable in terms of the discrete events on the replicator level. But the real innovation with the approach we take here is the study of the transitory dynamics that is feasible because the two different scenarios are algorithmically possible to link. We have inserted a few sentences explaining this into the Introduction (lines 169-181 in MA with track changes).

Another thing that could be addressed in more depth would be to mention other RNA replication computational/simulation models in the literature and to better contextualize the key added value(s) of combining both models. Virtually all of the references to the two models discussed are co-authored by the authors and their collaborators. What simulations from others have been done on these questions? As well as additional reference to the work of Szostak on protocellular evolution mentioned above, there is also no mention of other key contributors to questions of the OoL, e.g. Cech, Altman, Koonin, etc. Eigen is briefly mentioned but it would be good to elaborate more on his specific work in relation to OoL simulations. A more comprehensive review of the literature and situating the model and findings in the context of the wider field would be appropriate and useful. Because the computational aspects of the paper are difficult for the non-expert to follow, as it stands now, this paper is of value mostly to researchers with computational expertise. To be more useful to a broader audience with interest in questions of the Origin of Life, better transitioning between theoretical approaches and conclusions to expand on the literature and possible future approaches might be appropriate and valuable.

We have inserted some more references representing a few more contributions to the field. We are, however, not convinced that the Hypercycle model warrants many more mentions in the origin of life context, mainly because it has been shown many times that it is not capable of resisting its parasitic mutants (refs. #33, #34, #51, #52 and Scheuring, 2003, Szathmáry, 2013), besides it assuming specific direct help from one Hypercycle member to the other, which is not a realistic assumption regarding RNA or RNA-like replicators.

References not cited in manuscript:

- Scheuring, I. et al. Spatial models of prebiotic evolution: soup before pizza. *Origins of Life and Evolution of Biosphere* 33, 319-355 (2003).
- Szathmáry, E. On the propagation of a conceptual error concerning hypercycles and cooperation. *Journal of Systems Chemistry* 4, 1 (2013).

Response to Reviewers

Dear Reviewers,

Please find our replies to the reviews below in *italics*.

Yours sincerely,

Tamás Czárán
on behalf of all authors

Reviewer #1

I remain more skeptical than the authors of this paper regarding the role of mineral surfaces in the origin of cells, but the authors are entitled to their view, and the revision is acceptable.

Thank you for your comments and suggestions! We are aware that the role of mineral surfaces at the origin of life is still a hot and controversial topic, and it requires further discussion.

Reviewer #2/3

These comments reflect the response of this reviewer to the rebuttal by the authors to the initial review of the manuscript "The dynamics of prebiotic take-off: from mineral surfaces to vesicles" by Dr Czárán and colleagues, which has been revised for Communications Biology in response to previous comments. The authors present a theoretical analysis of two models of prebiotic evolution, MCRS and SCM, in the context of a transition from RNA replicator/ribozyme communities existing in 2D, specifically, on mineral surfaces, to protocells. They conclude that MCRS is a better model for the earliest stages of evolution, presumably, random RNA ensembles on mineral surface, whereas SCM works better at the protocellular stage, and that the transition between the two models is feasible. In their rebuttal, the authors present what we see as a minimal response to the major concerns of Reviewer 1.

While there are other hypotheses about the possible environment of the first living systems, we believe that mineral surfaces have the most theoretical and empirical support. We certainly did not mean to disparage other hypotheses; we just aimed at investigating the scenario for the transition to vesicles that seems the most plausible one, at least for us. Reviewing all the pros and cons of all the hypothetical environments is far beyond the scope and extent of this manuscript. Other than that, we have tried to build all the comments of Reviewer #1 into this revision.

The rebuttal focuses on a mineral surface and little else. The authors did much else so this essentially exclusive focus on the issue of whether a mineral surface is 'feasible and necessary' may still be overstated. It is a model so it may not be the most relevant if it is 'necessary'. The rebuttal response still appears to be overstated. A better response may be of the form 'it is not absolutely necessary but it provides a useful theoretical framework for thinking about the question. A theory is to inform thinking, understanding and future questions, not to determine what 'must' have occurred.

The authors correctly respond that a joint mechanism of compartmentalization and catalysis is useful, since parallel origins of the two processes is less plausible. They then however conclude that a mineral surface *must* be the solution. There are clearly other possible models, some already proposed and others under intensive present investigation that can occur. A better way to frame this might be that the mineral surface provides one possible solution but it is not *the* solution.

Thank you for your comments, we have edited the corresponding parts; we hope that the text is more accurate now.

Regarding comments of Reviewer 2, the authors did not seem to sufficiently address the first paragraph in the review. They mention adding three lines to address the study by Gozen that was “to a large extent, our motivation for this theoretical approach”. They have addressed this now but more than a 3-sentence response would be justified for the ~2 paragraphs of feedback provided by Reviewer 2.

We have modified the manuscript along the lines suggested. Since the article is already quite long, we kept the inserted pieces of text as brief as possible.

The authors note that the ‘real innovation’ was “the study of the transitory dynamics that is feasible because the two different scenarios are algorithmically possible to link.” This is now partially clarified, but it is not clear that the models implemented were already in existence. This has now been highlighted in lines 169-181 which is useful. In this context, the work presented in the manuscript is not unique because the models implemented were already in existence by the authors themselves and by others. Nonetheless, this work does advance knowledge about this question and publication is warranted.

In thinking about the ‘necessity’ of mineral surfaces, this model can be useful for thinking about the origin of life even if mineral surfaces were not the mode of pre-membrane compartmentalization. The authors can still make a case for their model without overstating its ‘necessity’.

We have modified the manuscript; the current version of the Introduction and Discussion is now less focused exclusively on mineral surfaces.

It seems a major motivation of the study is to explore the dynamics of the '2D to 3D' replication surface, i.e., replication on a 'flat' mineral surface evolving into a 3D space in which replication occurs. This appears more strongly emphasized in the present manuscript version provided compared to the initial submission. But due to the manuscript's overly verbose style and somewhat lack of a coherent and clear message, the writing needs additional strong editing

The authors have added some of the references suggested by Reviewer 2 but are still missing some key references, such as Cech and Altman's early 1980s papers, and more recent work from Koonin.

We have included some references suggested by the Reviewer.

The tracked/edited manuscript provided to the reviewers is very difficult to follow and needs extensive editing by the journal copy editors which is beyond the scope of this reviewer. The grammar and writing style require significant editorial improvement. As well as standard editorial improvement, a clear, strong, and supportable conclusion of the work would help.